# Individual differences in alexithymia modulate cognition-emotion interactions in daily life ongoing experiences
Anqi Lei [1,2], Md Faysal [1,2], Louis Chitiz[3], Raven Wallace[3], Samyogita Hardikar [3], Brontë McKeown [3], Jonathan Smallwood [3], Elizabeth Jefferies [4], Robert Leech [5] & Nerissa Ho [1,2] ✉

Individual differences modulate our thoughts and emotional experiences, yet how thought and emotion interact in daily life remains largely unclear. We leverage alexithymia, a trait reflecting atypical emotional awareness, to reveal these interactions in naturalistic settings. Using multi-dimensional experience sampling via smartphones, we captured moment-to-moment thought patterns and concurrent affective states (valence, arousal, stress) in people's daily life ($N$ = 190 undergraduate students, age range = 18 to 36, 159 females). Using Principal Component Analysis and Linear Mixed Models, we identified four thought dimensions that relate differently to these affective states: future-self orientation, intrusive distraction, sensory engagement, and task-focus. Alexithymia modulated these relationships. High overall alexithymia predicted fewer future-self-oriented thoughts and greater variability in sensory engagement across affective and social contexts, while difficulty identifying feelings selectively reduced future-self orientation during intense sadness, and externally oriented thinking rendered thought patterns less sensitive to affective context. By mapping affective experiences onto thought dimensions, these findings uncover cognitive pathways that connect to emotional well-being, providing a scalable framework for understanding variability in human affective experience.

Thought patterns and emotional experience are intimately intertwined. In daily life, a substantial proportion of self-generated thought carries an affective tone[1]. Maladaptive thought patterns are common in emotional disorders such as depression, anxiety, and post-traumatic stress[2–4], with strong evidence that repetitive negative thinking (rumination and worry) sustains and exacerbates negative mood[5,6]. Therapeutic approaches such as cognitive-behavioural and mindfulness-based interventions target these thought patterns to improve emotional well-being, highlighting the functional importance of thought-affect interactions[7,8]. Beyond clinical constructs, cognitive models of emotion, particularly appraisal theories, also provide a broader foundation for understanding thought-affect interactions[9,10]. These theories propose that cognitive evaluations of external and internal events along different dimensions (e.g., goal congruence, importance, agency, intentionality, coping potential) predict the quality of emotional experience and core affective dimensions, including valence and arousal[11,12]. Complementing these accounts, recent research on self-generated thought shows that negative affect, particularly sad mood, is consistently associated with intrusive, past-focused mind wandering[13–15], whereas positive affect tends to accompany task-related, prospective and socially oriented thought patterns[16,17]. This thought-affect relationship has been formalised in the content regulation hypothesis[18,19], which suggests that the affective outcomes of task-unrelated thoughts are not always negative, but depend on their experiential contents.

Despite this extensive investigation of cognition-emotion interactions, much existing work relies on controlled laboratory tasks or retrospective self-reports that are prone to memory biases. To improve ecological validity, real-life experience sampling methods are increasingly used to characterise the relationship between thought patterns and affective valence[17,20,21]. However, most of these studies focused on specific sets of thought (e.g., mind wandering, rumination), limiting a comprehensive understanding of how heterogeneous thought patterns map onto broader affective experiences in daily life. Moreover, beyond valence, high arousal has been linked to increased self-related contemplation and task-focus, whereas low arousal is associated with inattentive or off-task states[22,23]. Subjective stress (i.e.,

[1]School of Psychology, Faculty of Health, University of Plymouth, Plymouth, UK. [2]Brain Research & Imaging Centre, University of Plymouth, Plymouth, UK. [3]Department of Psychology, Queens University, Kingston, ON, Canada. [4]Department of Psychology, University of York, York, UK. [5]Institute of Psychiatry, Psychology & Neuroscience, King's College, London, UK. ✉e-mail: nerissa.ho@plymouth.ac.uk

personal perception of stress), though less frequently studied, also shapes cognition, influencing thought patterns depending on context and potentially modulating adaptive or maladaptive coping[24,25]. These findings collectively motivate the inclusion of valence, arousal, and stress as key affective dimensions in our study.

A further gap concerns how individual differences in emotion-related traits modulate thought-affect interactions. In the current study, we adopt alexithymia[26], a multi-faceted personality trait reflecting atypical emotional awareness, as the construct for examining individual differences in thought-affect interactions. This is theoretically relevant because alexithymia is composed of three sub-facets that are conceptually defined by different disruptions in the integration of cognitive and emotional processes[27]: difficulty identifying feelings (DIF), difficulty describing feelings (DDF), and externally oriented thinking (EOT). The DIF facet captures the degree of difficulty in recognising one's feelings and distinguishing them from physical sensations, the DDF facet reflects difficulty in verbalising one's feelings to other people, and the EOT facet describes the tendency to focus on external events rather than internal feelings[28]. Based on the attention–appraisal model of alexithymia[29], people with high alexithymia have reduced attention to emotional cues (high EOT) and/or impaired cognitive appraisal (high DIF/DDF). The three-process model of emotional awareness[30,31] also proposed that alexithymia involves disruptions in accessing and communicating cognitive meanings of emotions (high DIF/DDF), detecting affective signals for setting cognitive priority (high EOT), and applying cognitive control over emotional states through attention and working memory (all three sub-facets). Yet, how these facets connect to momentary thought–affect relationships remains largely unexplored in naturalistic, daily life contexts.

To address these issues, we employed multi-dimensional experience sampling (MDES) via smartphones to capture moment-to-moment thoughts and affective states in daily life[32]. Participants reported their thoughts along 18 dimensions, including temporal focus, self-relatedness, social orientation, task focus and intrusiveness, seven times per day over five days. Concurrently, they rated affective states (valence, arousal, and stress) and social contexts (alone or with others). Using Principal Component Analysis, we identified four latent dimensions of on-going thought: future-self orientation, intrusive distraction, sensory engagement, and task-focus. MDES has been validated both in the laboratory, where it tracks dynamic changes in brain activity and reproduces task-related brain organisation[33,34], and in daily life, where it reliably distinguishes thought patterns across activities and is sensitive to trait variation[35,36]. Linear Mixed Models (LMMs) were then employed to examine how affective states mapped onto these thought dimensions and the extent to which alexithymia traits modulated these relationships while accounting for the nested structure of the data.

Our study had three primary aims. First, to map state-level affective experiences onto patterns of ongoing thought in daily life. Second, to examine how alexithymia traits relate to variation in these thought patterns. Third, to determine whether alexithymia modulates the relationship between affective states and thoughts, or thoughts across social and solitary contexts. We hypothesised that different ongoing thought patterns would be linked with distinct affective states, and individuals with higher alexithymia would report less internally reflective, self-related, and detailed thought patterns, based on evidence that they tend to engage in more concrete, less introspective cognition, with greater experiential avoidance and use of emotional suppression[28,37,38]. Moreover, specific alexithymia facets were expected to modulate thought-affect dynamics: difficulty identifying or describing feelings (DIF, DDF) was predicted to disrupt thought regulation during negative affect[31], whereas externally oriented thinking (EOT) was expected to maintain focus on external stimulations rather than internal experiences[39]. Further, as general psychological distress can influence both affect and thought independently of alexithymia, we controlled for baseline distress using the 21-item Depression, Anxiety, and Stress Scale (DASS-21)[40]. This enables us to isolate the contribution of alexithymia traits from distress-related factors that may also moderate thought-affect relationships. Overall, by combining ecologically valid experience sampling with trait-level

measures of alexithymia, this study provides a framework for understanding how cognition and emotion dynamically interact in daily life, highlighting cognitive pathways underlying affective experience and trait-driven variability in emotional well-being, with relevance for both basic research and interventions targeting maladaptive thought-affect patterns.

## Methods
### Participants
Two hundred and twenty-five undergraduate students at the University of Plymouth participated in the daily-life experience sampling study, recruited through the SONA point system. Data were collected during two time periods: March to April 2023, and February to April 2024. The study was approved by the ethical committee at the University of Plymouth, and participants gave written informed consent online before taking part. Upon completion, participants were fully debriefed and received either a £10 Amazon voucher or two participation credits as compensation for their time and effort. Preregistration was not conducted for this study.

Thirty-five participants, in total, were excluded from the collected dataset for several reasons. These include providing problematic responses to the personality questionnaires (finished the questionnaires too fast, i.e., under 3 mins, $N = 3$; or gave the same response across all items, $N = 1$), receiving doubled probes per day due to a scheduling mistake ($N = 7$), responding to the thought probes at lower than the target response rate of 50% ($N = 22$), and submitting too many duplicate entries (more than 5 times) with different responses throughout their participation ($N = 2$).

After excluding the above participants, we applied the following exclusion criteria to the remaining observations in the dataset: duplicate entries for the same thought probe (i.e. only the first submission was retained); entries with identical ratings across all thought questions; entries with response times more than 3 standard deviations away from the mean; very fast responses ( < 30 seconds), which are not captured by the standard deviation exclusion criterion due to the right-skewed distribution of response times. These steps removed another 456 observations from the remaining dataset (8.15%). The final dataset consists of 5063 observations from 190 participants with an average age of 20.2 years (SD = 2.8 years, age range = 18 to 36), including 159 females, 28 males, and 3 who reported a non-binary gender (based on self-reported age and gender). We did not collect data on race/ethnicity or socioeconomic status for this study.

### State and trait measures
**Multidimensional Experience Sampling (MDES) Questionnaires.** Thoughts were measured by 18 thought questions in the MDES using a 0 to 10 Likert scale (0: not at all, 10: completely). A detailed description of each thought item is provided in Supplementary Table 3.

**Measures of affective states and other contextual information.** Affective states were rated by valence ("I was very happy", 0 = very sad, 10 = very happy), arousal ("My feeling was very strong", 0 = Not at all, 10 = Completely) and stress ("I was very stressed", 0 = Not at all, 10 = Completely) in response to the question prompt "How were you feeling?". It should be noted that the wording of the arousal statement in terms of "strong" was chosen to avoid potential confusion with jargon (e.g., "activated", "aroused"), and responses may reflect a mixture of perceived emotional strength and bodily activation. Distribution of ratings on affective states of valence, arousal, and stress is presented as histograms in Fig. 1C.

In addition, in order to understand the contextual situation in which thoughts and emotions were experienced, two categorical variables were also collected, namely a 5-option social environment: "Alone by myself, both physically and virtually", "Physically around people and interacting", "Physically around people but not interacting", "Virtually around people and interacting", "Virtually around people but not interacting". Social environments were recoded into two levels for later analysis: Alone (2118 observations) and in a social setting (2945 observations). Finally, from a list of activities and locations, participants selected one or more activities they

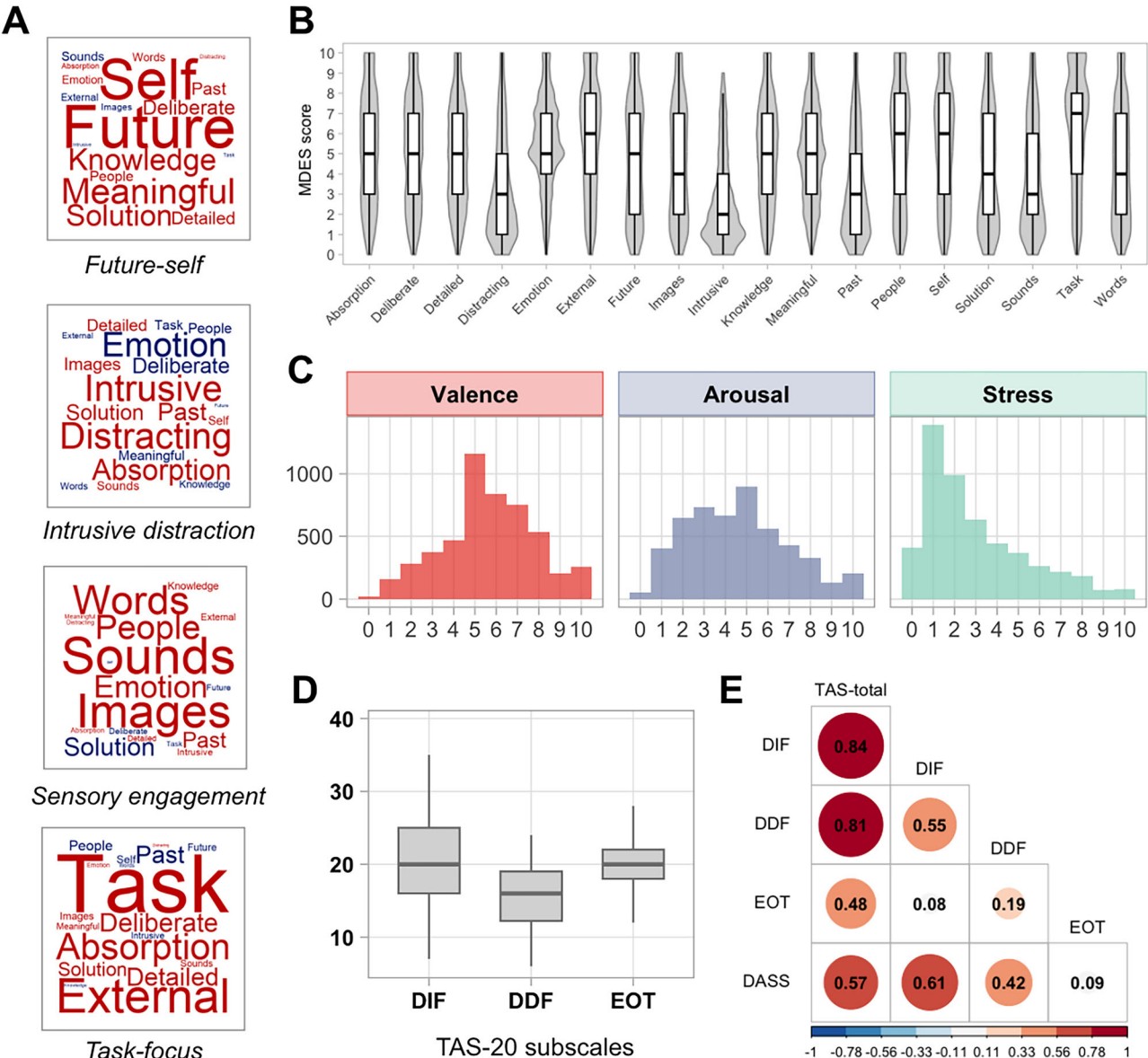

**Fig. 1 | An overview of PCA-derived dimensional structure of ongoing thought, mean rating on each thought item, and measures of affective states and alexithymia traits ($N$ = 190 participants). A** Word clouds visualising the thought item loadings for the 4 thought patterns identified using PCA: future-self (Component 1), intrusive distraction (Component 2), sensory engagement (Component 3), and task-focus (Component 4). Words in red indicate positive loadings, and negative loadings are in blue. Font size reflects the item loading magnitude. **B** Boxplots and violin plots showing the distribution of MDES scores for each dimension of thought. Scores could range from 0 (not at all) to 10 (completely). **C** Histograms showing the distribution of ratings on Valence, Arousal, and Stress (0-10 scale), with Valence rated from "very sad" to "very happy". **D** Boxplots showing the distribution of scores on the three TAS-20 subscales representing different core facets of alexithymia: DIF (Difficulty Identifying Feelings), DDF (Difficulty Describing Feelings), and EOT (Externally Oriented Thinking). **E** Correlation matrix showing Pearson's correlation coefficients between TAS-20 measures of alexithymia traits (TAS-total, TAS-DIF, TAS-DDF, TAS-EOT) and baseline distress as indexed by DASS scores (DASS = Depression, Anxiety, and Stress Scale). The strength and direction of correlations are indicated by size and colour, with numerical values representing exact coefficients. Note: For all boxplots, boxes show the median and interquartile range (IQR), and whiskers extend to 1.5× IQR.

engaged in and indicated their location when experiencing the thoughts and emotions (analyses on activity and location are not included as they are not the focus of our study).

**Measures of alexithymia.** The 20-item version of the Toronto-Alexithymia Scale[41] is a self-report questionnaire including three subscales: (1) Difficulty Identifying Feelings (DIF, 7 items), (2) Difficulty Describing Feelings (DDF, 5 items), and (3) Externally Oriented Thinking (EOT, 8 items). TAS-20 is among the most commonly used measures of alexithymia that has consistently demonstrated great validity and test-retest reliability in many languages and cultures[42]. Distribution of TAS-20 scores is shown as boxplots in Fig. 1D.

To determine whether our findings were confounded by potential behavioural biases—such as careless or restricted responding—that may be associated with alexithymia, we examined several established measures for identifying abnormal response patterns in experience sampling research[43]. As shown in Supplementary Table 2, these descriptive indices were largely comparable across low and high alexithymia levels.

**Baseline distress.** The 21-item version of the Depression, Anxiety, Stress Scale[40] was used to measure the negative emotional states of depression (7 items), anxiety (7 items), and stress (7 items) over the past week, using a 4-point Likert scale (0 = *never*, 3 = *almost always*). We used the total DASS score as a control of general baseline distress, as it often co-

occurs with alexithymia and might confound our results[44]. A descriptive summary of alexithymia-related measures and baseline distress can be found in Supplementary Table 1, and Pearson's correlation coefficients between TAS-20 measures (TAS-total and subscales) and DASS scores are shown in Fig. 1E.

## Procedure

The 5-day experience sampling period started from Friday morning to Tuesday night to ensure we could capture a diverse range of contexts from both working days and weekends. Participants received notifications from the Samply app[45] on their smartphones directing them to complete an experience sampling questionnaire on Qualtrics seven times a day at quasi-random intervals covering the entire day (2 probes between 9:00 to 12:30, 3 probes between 12:30 and 17:30, and 2 probes between 17:30 and 21:00), with at least one hour between probes. Participants were required to report their thoughts that occurred immediately before the notification, alongside their affective states, the social environment, where they were and what they were doing. They were informed that they needed to respond to at least ~75% of the probes in order to receive the course credit or payment. After the 5-day sampling, participants who reached the completion rate criteria were debriefed, and those who completed below the required rate were invited one more time for additional experience sampling days to catch up for the missing responses. Only participants who completed more than 50% of the probes after two invitations were included for analysis.

## Data analysis

**Principal component analysis (PCA).** To obtain a lower-dimensional space for the MDES data with minimal loss of crucial information, we applied PCA at the observation level with varimax rotation using the "ThoughtSpace" Python package (that utilises scikit-learn)[35]. Data were standardised, and outliers ($z>$ or $<3$) were replaced with mean values (0.08% corrected) before input for analysis. The Kaiser-Meyer-Olkin (KMO) measure of sampling adequacy was 0.85, and Bartlett's test of Sphericity was significant ($\chi^2(153) = 18784.01$, $p < 0.001$), indicating great suitability of the data for PCA. Components with eigenvalue greater than 1 were selected as distinct thought dimensions to be used for further analyses (see Supplementary Table 4 for the scree plot of eigenvalues). Individual component scores for each observation in the data were obtained by calculating the dot product between the component loadings of each thought dimension and the original ratings in the MDES dataset. Finally, to ensure the stability of thought dimension structures, we performed a bootstrapped split-half reliability with 1000 iterations using the *RHom* module in the "ThoughtSpace" package. This involved randomly splitting the data into two halves, running PCA separately on each half, correlating the component loadings, repeating this process 1000 times and taking the average correlation across all iterations (with 95% confidence intervals).

**Linear mixed effects models (LMMs).** We analysed the experience-sampling data (PCA scores) using LMMs with random intercepts for participants to account for the nested structure of observations within individuals[46]. Using the *lme4* and the *lmerTest* packages[47,48] in R (Version 2024.04.2 + 764), we fitted LMMs using restricted maximum-likelihood estimation (REML) to estimate fixed effects and obtain p-values from *F*-tests and *t*-tests. The Satterthwaite approximation was used to calculate degrees of freedom. When categorical predictors (i.e., social environments) were involved, we applied the "contr.sum" contrasts to compare various conditions to the grand mean of the conditions. All numeric variables were centred and scaled prior to the LMM analysis. Moreover, for each LMM, age and gender were included as covariates while participant ID was included as a random intercept. Normality assumptions of the LMMs were checked via visual inspection of Q–Q plots of the residuals, which indicated no substantial violations.

Results for each set of LMM were Bonferroni corrected at $p < 0.0125$ to account for multiple comparisons of the four dimensions of thought.

Significant interactions were plotted to visually examine the effects, followed by post-hoc comparisons (including contrasts of contrasts) using the *emmeans* package[49], with significance thresholds Bonferroni-adjusted based on the number of tests. When probing interactions involving a continuous predictor, we defined "low" and "high" levels using the minimum and maximum values of that predictor. Effect sizes for fixed effects were reported using partial $R^2$, and overall model fit was summarised as marginal ($R^2m$) and conditional ($R^2c$) R-squared for LMMs.

**The effect of affective states on thoughts.** To understand the patterns of thoughts in relation to affective experiences, we modelled the main effect of affective states (valence, arousal, and stress) and the interaction between valence and arousal on the component scores for each of the four thought dimensions separately. We included Social Environment as a covariate to control for the effect of social settings as well as reveal its possible influence. An example formula is as follows: lmer (Thought ~ Valence * Arousal + Stress + Age + Gender + Social Environment + (1| Participant ID)).

**The effect of alexithymia-related traits on thoughts.** Next, we modelled LMMs with TAS-20 measures (total and subscale scores) as main predictors for 4 dimensions of thoughts while controlling for DASS-total score, age, and gender. An example formula is as follows: lmer(Thought ~ TAS-20 measure(s) + DASS + Age + Gender + (1| Participant ID)).

**Interaction effects of alexithymia-related traits and social-affective contexts on thought patterns.** Lastly, we performed a set of LMMs with TAS-20 measures and their interaction with the affective variables outlined above, while controlling for DASS scores, age, gender, and social environment using the formula: lmer(Thought ~ (Valence * Arousal + Stress) * TAS-20 measure(s) + Social Environment + DASS + Age + Gender + (1|Participant ID)). Interactions between TAS-20 measures and social environments were tested using the formula: lmer(Thought ~ Social Environment*TAS-20 measure(s) + DASS + Age + Gender + (1|Participant ID)).

## Results

### Dimensional structure of on-going thought in daily life

To identify the dimensional structure of daily life thought patterns, consistent with prior studies[32,35], we applied Principal Components Analysis (PCA) to the MDES data to generate a simplified set of dimensions that best explained self-reported descriptions of thinking in daily life. Four components (eigenvalue > 1) that accounted for ~50% of the total variance were selected (see Methods section and Supplementary Tables 4 and 5 for details). Thought components were labelled based on the most highly loaded features from both positive and negative directions, with negative variables interpreted as representing the opposite meanings of the corresponding features.

Component 1, labelled as "Future-Self", described a pattern of future-focused thoughts involving the self that accounted for 23.58% of the total variance and loaded highly on "Future", "Self", and "Meaningful" features. Prior studies have found that this pattern of thought is related to the process by which people generate more concrete personal goals[50] and can be important in the resolution of unhappy mood states[16]. Component 2, named as "Intrusive Distraction", described a pattern of unpleasant distracting thought that accounted for 11.54% of the total variance and loaded positively on "Intrusive", "Distracting" but negatively on "Emotion". Prior studies have shown that this aspect of thought is associated with worse performance on laboratory tasks[51], impaired memory and moments characterised by reduced executive control[34]. Component 3, termed "Sensory Engagement", contributed 8.45% of the total variance and was characterised by "Sounds", "Images", and "Words". In the laboratory, this pattern of thought is linked to the engagement of sensory systems during movie watching[34]. Component 4, labelled as "Task-focus", explained 5.84% of the

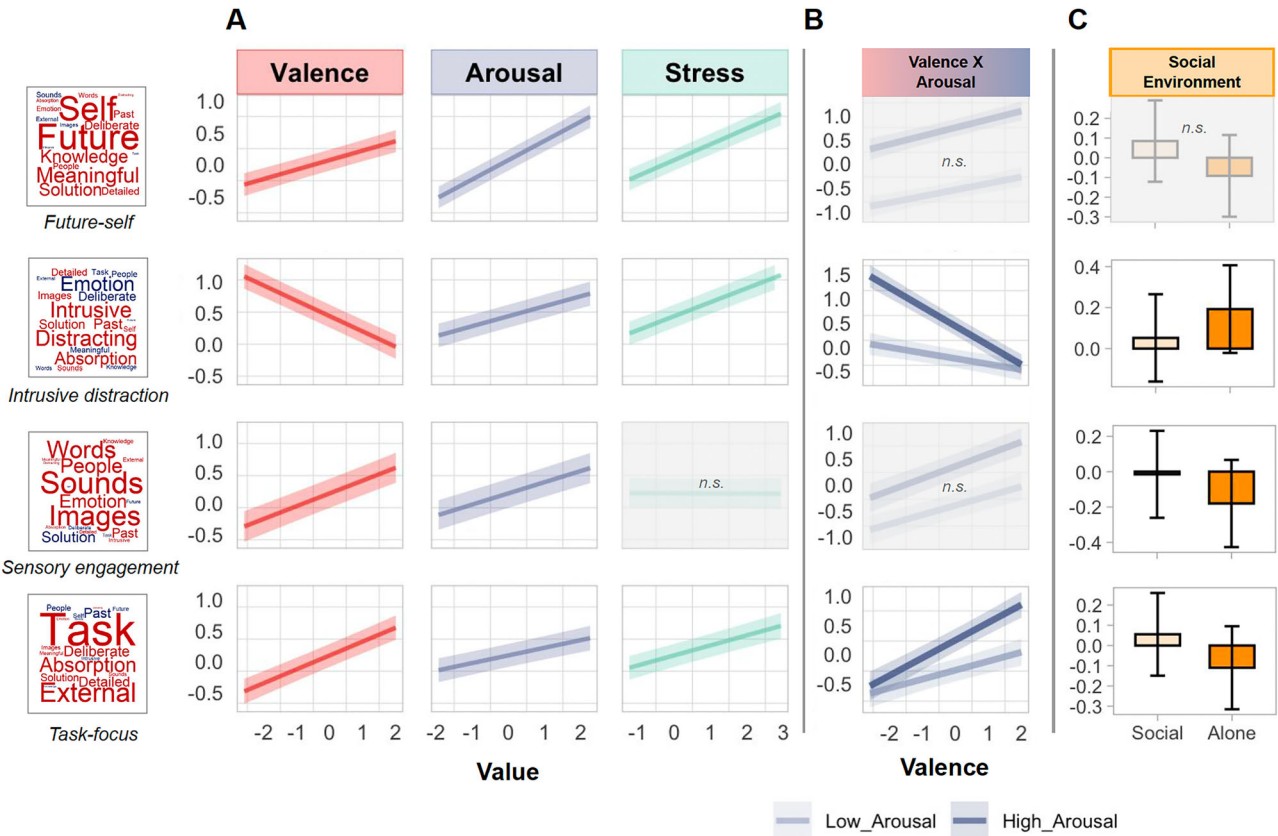

**Fig. 2 | Effects of affective states and social environment on four thought dimensions, presented as estimated marginal means (EMMs) with 95% confidence intervals (CIs) ($N = 190$ participants).** Rows correspond to the four dimensions of thought: (1) future-self, (2) intrusive distraction, (3) sensory engagement, and (4) task-focus. **A** Main effects of z-scored Valence (red), Arousal (purple), and Stress (green) on each thought dimension. Lines are model-predicted EMMs; shaded ribbons are 95% CIs. The x-axis shows the value of each predictor.

**B** Two-way Valence × Arousal interactions for intrusive distraction and task-focus. The x-axis corresponds to the value of Valence (ranging from sad to happy), and the line colour represents two levels of Arousal. Dark purple = high Arousal; light purple = low Arousal. **C** Main effect of Social Environment on each thought dimension: bars show EMMs ( ± 95% CI) for Social (pale orange) versus Alone (dark orange). Shaded panels labelled "n.s." did not reach significance.

total variance and loaded highly on "Task", "External", and "Absorption". This pattern of thought emerges in complex laboratory tasks that rely on executive control[52], and neuroimaging studies have linked it to activity in the brain's executive control system[53]. Figure 1A visualises the loadings of thought items on each component as a word cloud, and Fig. 1B plots the distribution of ratings on each thought item. Bootstrapped split-half reliability analysis, based on 1000 iterations, revealed strong internal reliability of the PCA, with an average homologue similarity score of 0.992 (95% CI [0.985, 1]).

**How do affective states map onto thought patterns?**
Having identified the four dimensions of daily life thoughts, we first examined whether these thought dimensions are related to different affective states. The results revealed significant main effects of Valence and Arousal on all four thought dimensions (all $p < 0.001$, see Supplementary Table 6 for $F$-statistics and Fig. 2A for visualisation). Regarding Valence, a happier mood was positively associated with future-self orientation ($\beta = 0.15$, 95%CI [0.12, 0.18], $p < 0.001$, R²m/R²c = 0.24/0.39), sensory engagement ($\beta = 0.20$, 95%CI [0.17, 0.23], $p < 0.001$, R²m/R²c = 0.09/0.39), and task-focus ($\beta = 0.22$, 95%CI [0.19, 0.25], $p < 0.001$, R²m/R²c = 0.10/0.27), but negatively associated with intrusive distraction ($\beta = -0.24$, 95%CI [−0.27, −0.21], $p < 0.001$, R²m/R²c = 0.23/0.42). Arousal positively predicted all dimensions of thought (all $p < 0.001$), meaning that higher arousal was associated with increased engagement across all types of thought, while lower arousal would correspond to the opposite ends of these thought dimensions such as "pleasant deliberate" thoughts (the opposite pole of "intrusive distraction") and "off-task internally focused"

thoughts (the opposite pole of "task-focus"). Association between lower arousal and decreased task-relevant focus is consistent with prior research[22,23]. Similarly, stress positively predicted all dimensions of thought (all $p < 0.001$), except for sensory engagement ($\beta = -0.002$, 95%CI [−0.03, 0.03], $p = 0.873$).

Significant interaction effects between Valence and Arousal were also observed for intrusive distraction ($F(1,5028.23) = 42.96$, $p < 0.001$, partR² = 0.004 [95%CI: 0, 0.027]) and task-focus ($F(1,5047.64) = 9.28$, $p = 0.002$, partR² = 0.003 [95%CI: 0, 0.034]) (See Fig. 2B). Compared to mild feelings (low arousal), strong affective feelings (high arousal) were linked to more intrusive distraction when people were sad ($\beta = -0.07$, 95%CI [−0.09, −0.05], $p < 0.001$), and more task-focused thoughts when they were happy ($\beta = 0.03$, 95%CI [0.01, 0.06], $p = 0.002$). This demonstrates that the effects of arousal were not uniform across these thought dimensions but depended on the emotional valence of the affective states. No significant interaction effects were observed for future-self focus ($F(1,5047.47) = 0.79$, $p = 0.376$, partR² = 0.0007 [95%CI: 0, 0.033]) and for sensory engagement ($F(1,4993.456) = 1.44$, $p = 0.231$, partR² = 0.0006 [95% CI: 0, 0.028]).

Finally, there was a significant main effect of Social Environment on all dimensions of thought (all $p < 0.0125$; see Fig. 2C), except for future-self ($\beta = 0.02$, 95%CI [0.00, 0.05], $p = 0.039$). Being in a social setting was associated with more sensory engagement ($\beta = 0.04$, 95%CI [0.02, 0.07], $p = 0.001$) and task focus ($\beta = 0.03$, 95%CI [0.01, 0.06], $p = 0.011$), but less intrusive distraction ($\beta = -0.07$, 95%CI [−0.09, −0.04], $p < 0.001$) compared to being alone. Reduction in intrusive distraction during social situations replicates prior studies[32,36].

## High alexithymia predicts reduced future-oriented self-focus in thinking

Having mapped the thought dimensions with various affective states, our next step was to examine if such relationships would be moderated by dispositional emotional factors, specifically alexithymia traits. First, we explored whether overall alexithymia and its core facets would predict each of the four thought dimensions (see Supplementary Tables 7 and 8). LMMs results showed that TAS-total score had a main effect only on future-self ($F(1,184.86) = 6.55, p = 0.011$, partR$^2$ = 0.0093 [95%CI: 0, 0.058]), such that higher overall alexithymia was associated with reduced engagement in thoughts that are futuristic, self-oriented, meaningful, and solution-oriented ($\beta = -0.12$, 95%CI [−0.21, −0.03], $p = 0.010$, R$^2$m/R$^2$c = 0.05/0.28) (see Fig. 3A).

None of the alexithymia core facets (TAS-20 subscales) were significant predictors of any thought dimension after Bonferroni correction ('Future-self': DIF ($F(1,183.03) = 1.29, p = 0.257$, partR$^2$ = 0.0016 [95%CI: 0, 0.0496]), DDF ($F(1,183.29) = 4.41$, $p = 0.037$, partR$^2$ = 0.0062 [95%CI: 0, 0.054]), EOT ($F(1,182.44) = 0.08$, $p = 0.775$, partR$^2$ = 0.00 [95%CI: 0, 0.048]); 'Intrusive Distraction': DIF ($F(1,181.49) = 0.12, p = 0.729$, partR$^2$ = 0.0001 [95%CI: 0, 0.049]), DDF ($F(1,181.73) = 4.94$, $p = 0.027$, partR$^2$ = 0.0078 [95%CI: 0, 0.056]), EOT ($F(1,180.95) = 0.64$, $p = 0.424$, partR$^2$ = 0.0008 [95%CI: 0, 0.050]); 'Sensory Engagement': DIF ($F(1,182.00) = 0.69$, $p = 0.409$, partR$^2$ = 0.0014 [95%CI: 0.0002, 0.036]), DDF ($F(1,182.17) = 1.30, p = 0.256$, partR$^2$ = 0.0026 [95%CI: 0.001, 0.037]), EOT ($F(1,181.63) = 0.07$, $p = 0.799$, partR$^2$ = 0.0001 [95%CI: 0, 0.035]); 'Task-focus': DIF ($F(1,182.98) = 1.59$, $p = 0.209$, partR$^2$ = 0.0021 [95%CI: 0, 0.027]), DDF ($F(1,183.26) = 2.20$, $p = 0.139$, partR$^2$ = 0.003 [95%CI: 0, 0.028]), EOT ($F(1,182.36) = 0.08$, $p = 0.779$, partR$^2$ = 0.0001 [95%CI: 0, 0.025]); see Supplementary Table 8).

## Alexithymia-related traits modulate thought patterns depending on social-affective contexts

Having examined the separate effects of affective states and alexithymia traits on dimensions of thought, we next modelled interactions between TAS-20 measures and affective states as well as interactions between TAS-20 measures and social environments through a set of LMMs (see Supplementary Tables 9–12 for the detailed outputs from these models). Below are the findings for each thought dimension.

**Future-self**. There was a significant three-way interaction between TAS-DIF, Valence, and Arousal in predicting future-self related thoughts ($F(1,5019.01) = 7.15$, $p = 0.008$, partR$^2$ = 0.0011 [95%CI: 0, 0.048]). As plotted in Fig. 3B, this interaction was driven by valence-modulated differences in thought between people with higher and lower TAS-DIF scores under strong arousal ($\beta = 0.03$, 95%CI [0.01, 0.06], $p = 0.008$, R$^2$m/R$^2$c = 0.26/0.40). Post-hoc analysis revealed that, under strong arousal states, only people with more difficulty identifying feelings (i.e., higher TAS-DIF) showed significantly less future-self focus under a sad mood than a happy mood ($estimate = 1.68$, 95%CI [1.00, 2.36], $p < 0.0001$); conversely, under weak arousal, only people with a better ability to identify emotions (i.e., lower TAS-DIF) engaged more in future-self thoughts when they were in a happy mood than a sad mood ($estimate = 1.19$, 95%CI [0.51, 1.88], $p < 0.001$). Importantly, contrasts of contrasts in strong arousal situations revealed that the reduction in future-self thoughts during sad versus happy mood was much greater in people with more difficulty identifying feelings than those with a better ability (happy–sad: high TAS-DIF > low TAS-DIF, $estimate = 0.40$, 95% CI [0.13, 0.67], $p = 0.004$). This valence-modulated difference in low arousal situations was not significant ($estimate = -0.26$, 95%CI [−0.56, 0.04], $p = 0.10$).

**Intrusive distraction**. A significant three-way interaction was also observed for intrusive distraction between TAS-EOT score, Valence, and Arousal ($F(1,5037.96) = 6.69$, $p = 0.010$, partR$^2$ = 0.0008 [95%CI: 0, 0.038]). As shown in Fig. 3C, the interaction was driven by valence-

modulated differences in thought between people with higher and lower TAS-EOT scores under strong arousal ($\beta = 0.03$, 95%CI [0.01, 0.05], $p = 0.01$, R$^2$m/R$^2$c = 0.28/0.44). Post-hoc tests revealed that only people with a less externally oriented thinking style (i.e., lower TAS-EOT) showed more intrusive distraction during intense experience of sadness compared to happiness ($estimate = -2.87$, 95%CI [−3.47, −2.27], $p < 0.0001$), whereas people with a highly externally oriented thinking style (i.e., high TAS-EOT) did not ($estimate = 0.26$, 95%CI [−0.77, 1.30], $p = 0.62$). Contrasts of contrasts revealed that such valence-modulated change in intrusive distraction under strong arousal was significantly larger in people with a less externally oriented thinking style than those with a higher external orientation (happy–sad: high TAS-EOT < low TAS-EOT, $estimate = 0.68$, 95%CI [0.34, 1.02], $p = 0.0001$). Again, this valence-modulated difference in low arousal situations was not significant ($estimate = -0.13$, 95%CI [−0.50, 0.24], $p = 0.49$). This suggests that the amount of intrusive distractive thoughts in people with a highly externally oriented thinking style (high TAS-EOT) was relatively invariable and unlikely to be changed by affective valence in comparison to those with a more introspective tendency.

**Sensory engagement**. There was a significant two-way interaction between TAS-total score and Valence on the level of thoughts dominated by sensory information ($F(1,5009.59) = 6.75$, $p = 0.009$, partR$^2$ = 0.0001 [95%CI: 0, 0.036]). As shown in Fig. 3D, the interaction was driven by valence-modulated differences in thought between people with higher and lower TAS-total scores ($\beta = 0.04$, 95%CI [0.01, 0.06], $p = 0.009$, R$^2$m/R$^2$c = 0.11/0.39).

Post-hoc tests showed that although people with both high and low overall alexithymia reported more engagement of sensory information under happy than sad states ($estimate = -1.32$, 95%CI [−1.64, −0.99], $p < 0.0001$ and $estimate = -0.53$, 95%CI [−0.86, −0.21], $p = 0.001$, respectively), people with high overall alexithymia showed a stronger effect of Valence on sensory engagement compared to those with low overall alexithymia (happy – sad: high TAS-total > low TAS-total, $estimate = 0.78$, 95% CI [0.19, 1.38], $p = 0.009$).

In addition, we also observed a significant interaction between TAS-total score and Social Environment in predicting sensory engagement ($F(1,4994.84) = 7.10$, $p = 0.008$, partR$^2$ = 0.0009 [95%CI: 0, 0.031], see Supplementary Table 11), driven by differences in social environment-modulated changes in thought across people with high and low overall alexithymia ($\beta = 0.03$, 95%CI [0.01, 0.06]), $p = 0.008$, R$^2$m/R$^2$c = 0.02/0.36). Post-hoc tests showed that people with higher overall alexithymia showed significantly more thoughts focused on concrete sensory information (sounds, images) in social settings than when alone ($estimate = 0.33$, 95%CI [0.20, 0.46], $p < 0.0001$), but not those with lower overall alexithymia ($estimate = 0.004$, 95%CI [−0.12, 0.13], $p = 0.95$). This difference was significantly bigger for people with higher than lower overall alexithymia (social–alone: high TAS-total > low TAS-total, $estimate = 0.33$, 95%CI [0.09, 0.57], $p = 0.008$). No significant effects emerged for other TAS-20 subscales after Bonferroni correction (see Supplementary Table 12).

**Task-focus**. Significant two-way interaction effects were observed on task-focus for TAS-EOT with Valence ($F(1,5034.56) = 6.33$, $p = 0.012$, partR$^2$ = 0.0009 [95%CI: 0, 0.031]; $\beta = -0.04$, 95%CI [−0.06, −0.01]), Arousal ($F(1,5037.72) = 8.37$, $p = 0.004$, partR$^2$ = 0.0024 [95%CI: 0, 0.033]; $\beta = 0.04$, 95%CI [0.01, 0.07]), and Stress ($F(1,5037.47) = 10.08$, $p = 0.002$, partR$^2$ = 0.0024 [95%CI: 0, 0.033]; $\beta = -0.05$, 95%CI [−0.08, −0.02], R$^2$m/R$^2$c = 0.11/0.28) (see Fig. 3F).

Post-hoc comparisons revealed that people who have a less externally oriented thinking style (low TAS-EOT) showed more task-focus when feeling happy than feeling sad ($estimate = 1.44$, 95%CI [1.06, 1.82], $p < 0.0001$) and when feeling more stressful than calm ($estimate = 1.20$, 95% CI [0.84, 1.57], $p < 0.0001$), whereas these valence- and stress-modulated changes were not observed in people with a highly externally oriented thinking style (high TAS-EOT) (for valence: $estimate = 0.21$, 95%CI [−0.41,

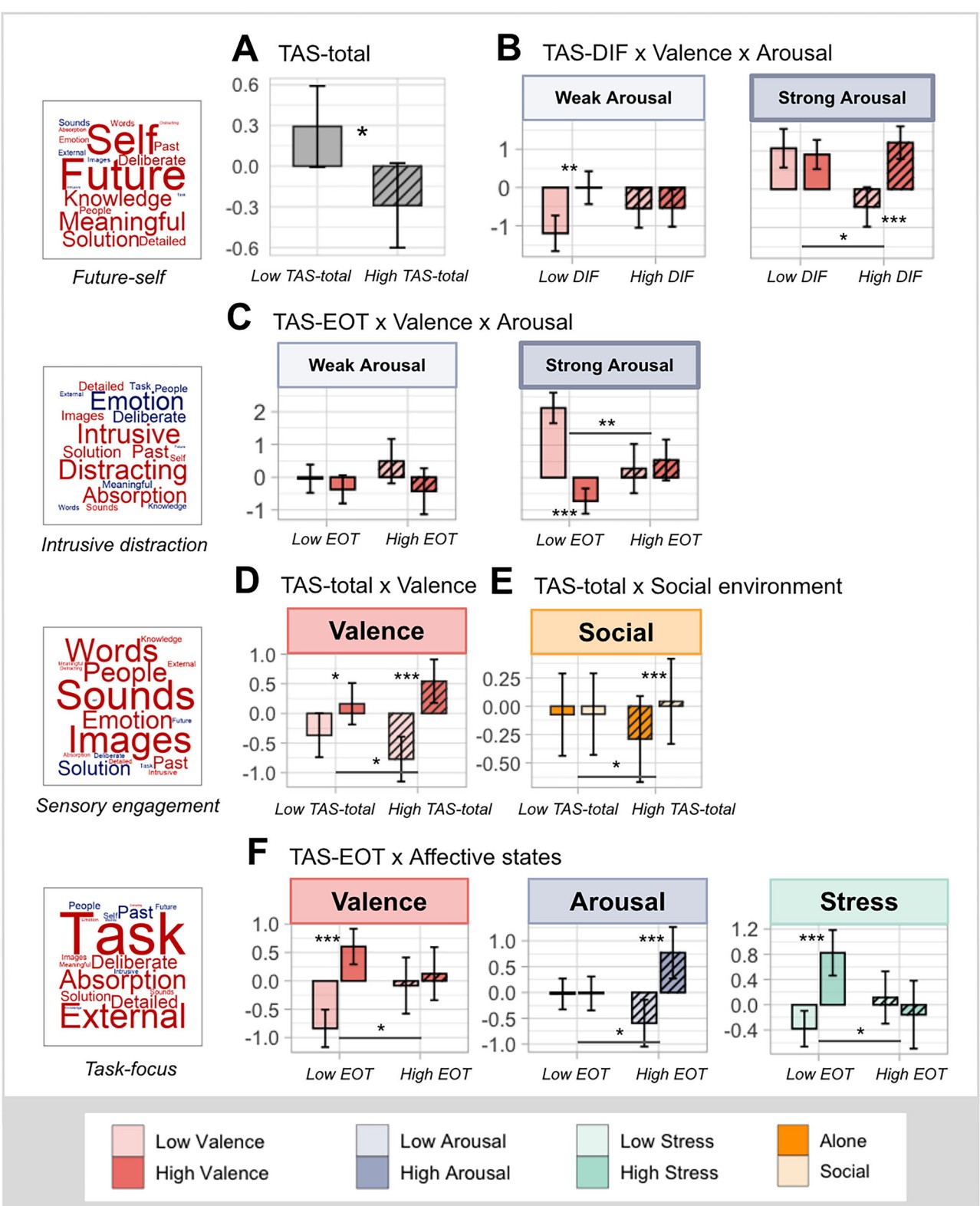

**Fig. 3 | Significant effects of TAS-20 measures and their interactions with affective states and social environment on thought dimensions ($N = 190$ participants).** Each row corresponds to results for the thought pattern shown on the left. Levels of TAS-related measures are depicted with solid (Low TAS) and striped fillings (High TAS), plotted on the left and right sides within each graph. Levels of affective predictors are depicted with light and dark shades (for Valence, light = sad, dark = happy). *Note:* "Low" and "High" levels of each continuous predictor correspond to the minimum and maximum observed values of that predictor. Significant results from post-hoc comparisons are indicated by *($p < 0.0125$), ** ($p < 0.001$),

and *** ($p < 0.0001$). Error bars represent 95% confidence intervals (CIs). **A** Main effect of TAS-total score on future-self thinking. **B** TAS-DIF × Valence × Arousal interaction on future-self orientation. **C** TAS-EOT × Valence × Arousal interaction on intrusive distraction. **D** TAS-total × Valence interaction on sensory engagement. **E** TAS-total × Social Environment interaction on sensory engagement. Levels of Social environments are depicted with pale orange (Social) and dark orange (Alone). **F** TAS-EOT × Valence, TAS-EOT × Arousal, and TAS-EOT × Stress interaction on task-focus.

0.83], $p = 0.51$; for stress: *estimate* $= -0.27$, 95%CI $[-0.86, 0.31]$, $p = 0.36$). However, people with a higher external orientation showed more task focus during strong arousal than weak arousal (*estimate* $= 1.37$, 95%CI $[0.76, 1.97]$, $p < 0.0001$), but not people with a lower external orientation (*estimate* $= 0.01$, 95%CI $[-0.34, 0.36]$, $p = 0.96$). Contrasts of contrasts showed that, compared to people with a less externally oriented thinking style, people with a highly externally oriented thinking style showed less valence-modulated (happy–sad: high TAS-EOT < low TAS-EOT, *estimate* $= -1.23$, 95%CI $[-2.19, -0.27]$, $p = 0.01$) and stress-modulated (high stress–low stress: high stress–low TAS-EOT, *estimate* $= -1.47$, 95%CI $[-2.39, -0.56]$, $p = 0.002$) changes in task-focus, whereas they showed more arousal-modulated changes in task-focus (high arousal – low arousal: high TAS-EOT > low TAS-EOT, *estimate* $= 1.36$, 95%CI $[0.44, 2.27]$, $p = 0.004$). This suggests that for people with a highly externally thinking style (high TAS-EOT), their task focus was relatively insensitive to the influence of valence and stress but more sensitive to the intensity of these feelings.

### Control analyses

Given the potential concern that people with higher alexithymia may have more limited access to their affective states, we also conducted analyses to examine whether alexithymia measures moderate affective ratings. Using total alexithymia scores and all three subscales to predict each momentary affect rating while controlling for age, gender, and DASS-total scores, we found no significant differences in reported valence, arousal, and stress across all alexithymia measures (all $p > 0.05$; see Supplementary Tables 13 and 14), except for a weak linkage of higher DDF to lower arousal ($\beta = -0.24$, 95%CI $[-0.46, -0.02]$, $p = 0.034$) and lower stress ($\beta = -0.25$, 95%CI $[-0.44, -0.05]$, $p = 0.015$). Hence, we consider that these results provide no strong evidence that alexithymia meaningfully biases momentary affect reports in our study.

Finally, to address the possibility that thoughts and affects might be confounded by the time of sampling, we reran the analyses, including the probe order (within each day) as an additional covariate in the LMMs. All key significant effects remained unchanged (see Supplementary Tables 15–19).

### Discussion

Our results provide direct evidence that the structure of daily life thoughts systematically relates to affective experience, and that these relationships are affected by dispositional differences in emotional awareness. By combining smartphone-based multidimensional experience sampling with measures of alexithymia, we show that individual facets of alexithymia uniquely influence how thought and affect interact in natural contexts. These findings highlight cognitive pathways that underpin emotional well-being and suggest novel mechanisms through which emotion-related traits confer risk for maladaptive outcomes.

We first established that distinct dimensions of thought are differentially associated with affective states. Future-self-oriented thinking—i.e., deliberate, goal-directed thoughts about one's future—was linked to greater happiness, arousal, and stress. This suggests that prospective thought simultaneously supports positive motivation and adaptive stress responses[16,25,54]. This also aligns with theories of prospective self-related simulation, which propose that imagining one's future self supports self-regulation and goal pursuit[55,56], and with findings from the cognitive appraisal literature linking positive affect to favourable evaluations of future events with better coping potential[12]. Task-focused thought was enhanced during highly aroused happy states and stress, consistent with evidence that engaging in the present can stabilise mood and performance[14,25]. Intrusive distraction, in turn, tracked stress and sadness, particularly under high arousal and in solitude, corroborating prior work linking repetitive negative thinking to negative mood and increased symptoms of depression[5,52,57]. Finally, sensory engagement, representing the emphasis on the modality of thoughts (e.g., sounds, images), was associated with happiness, arousal, and being in a social setting, consistent with research showing that multisensory stimulation and auditory attention amplify positive affect[58,59]. Interestingly,

sensory engagement was the only thought dimension not predicted by stress, despite some prior work linking heightened sensory processing to increased stress[60]. One explanation is that this kind of sensory engagement captured in our study differs qualitatively from the sensitivity to external and internal senses typically associated with stress, especially heightened interoception[61]. Together, these results demonstrate that daily-life thought is structured along functionally meaningful dimensions that map onto distinct affective experiences.

Building on these associations, we showed that alexithymia traits selectively modulate thought-affect dynamics. Individuals with higher overall alexithymia reported fewer future-self-oriented thoughts. Neuroimaging studies of alexithymia suggest reduced recruitment of brain regions supporting episodic simulation and self-related awareness[62,63], which may underlie this reduced future-self focus. Importantly, the reduction was most pronounced for individuals with difficulty identifying feelings (DIF) during episodes of intense sadness. This negativity-specific effect aligns with evidence that alexithymia involves negativity-avoidance strategies[64,65], such as suppression of distressing feelings, which may impair resilience by limiting the use of negative affect as a cue for adaptive coping. Notably, among the different facets of alexithymia, DIF has been shown as the best predictor of psychopathological symptoms in depression and anxiety[44] and is strongly implicated in emotion dysregulation[66].

We also found that individuals high in alexithymia showed stronger reliance on sensory engagement when happy and in social settings, but reduced sensory-focused thoughts when sad and alone. This supports qualitative reports that individuals high on alexithymia traits tend to describe joy in terms of concrete action and externally shared social events, rather than inner states[67]. In contrast, their reduced sensory engagement during solitude and sadness might reflect less self-generated multisensory simulation (recall or imagination of people-related events) when external social cues are not available. This reduced sensory engagement during sadness is also consistent with evidence that people with higher alexithymia show less vivid visual imagery when processing negative information, which is considered to hinder speedy emotion recognition[68].

Externally oriented thinking (EOT) further revealed how alexithymia traits modulate the flexibility of thought-affect coupling. Individuals with high EOT showed thought patterns less sensitive to valence and stress, with fewer shifts in intrusive distraction or task focus across affective states, suggesting that the flexibility of thought-affect dynamics depends upon the ability to adopt an introspective rather than an externally focused mindset. This reduced coupling may also reflect emotional dampening or detachment, limiting both distress and adaptive engagement with affective cues. Although blunted sensitivity might confer short-term protection against negative affect[69], it could also constrain opportunities for deep emotional processing, potentially contributing to rigid coping strategies and difficulties reflecting on emotions[65,70]. At the same time, people with high EOT maintained sensitivity to arousal, showing increased task focus during strong experiential states – consistent with the observation of alexithymia's reliance on external actions to regulate intense feelings or bodily sensations[71].

Together, these findings advance understanding of the cognitive mechanisms that link alexithymia to emotional difficulties. Rather than a global deficit, facets of alexithymia exert specific, context-dependent influences: DIF dampens prospective self-focused thought under intense sadness, EOT alters the dynamic coupling of task-relevant thought with affective context. This nuanced profile suggests that alexithymia alters not only emotional awareness but also the cognitive architecture that may support adaptive emotion regulation.

### Limitations

Several limitations warrant caution. Our measures of affective states might capture a mixture of chronic moods and transient episodes; future studies could disambiguate these temporal scales. We also note that although our PCA-driven thought patterns can identify higher-level structural dimensions of thoughts, we cannot determine the specific cognitive processes that

underlie each thought pattern nor the exact purpose or function of a given thought episode. Therefore, these thought patterns should be interpreted with caution. Future studies could enrich these interpretations, for example, by directly assessing the purpose of ongoing thoughts (e.g., goal-directed planning, emotion regulation, fantasy), or by combining experience sampling with think-aloud approaches[72] to measure more fine-grained content within each thought dimension. Additionally, future studies could benefit from asking participants to report on thoughts that are focused on internal bodily senses to better understand how interoceptive attention relates to affect, which may be distinct from that of visual or auditory senses[59], and thus, might be moderated by alexithymia in a different manner. Objective measures of affect (e.g., physiological methods) would ideally also be incorporated to complement subjective self-reports. Finally, our predominantly female, Western undergraduate sample limits generalisability; replication in more diverse and clinical populations will be crucial.

Despite these caveats, the present study demonstrates that sampling ongoing thought in daily life offers a powerful method for uncovering context-sensitive links between cognition and emotion. By situating individual differences in alexithymia within this framework, our findings highlight how dispositional traits influence the interplay of thought and emotion in ways that may underlie vulnerability to psychopathology. This integrative approach provides a scalable framework for mapping the cognitive pathways through which emotional traits influence well-being and offers new targets for interventions aimed at modifying maladaptive thought-affect dynamics.

## Data availability
The dataset for replicating the analyses in this article is publicly available from the Open Science Framework (OSF)[73]: https://doi.org/10.17605/OSF.IO/ZAVQF.

## Code availability
Our analyses relied upon Python (Version 3.11.7) and R (Version 2024.04.2 + 764). The Python code for replicating the PCA results in the article can be found at https://github.com/Bronte-Mckeown/ThoughtSpace. The R code for replicating the LMM analyses is publicly available at https://doi.org/10.17605/OSF.IO/ZAVQF.

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

## Acknowledgements

This work was supported by the BA/Leverhulme Small Research Grants SRG 2020-21 Round awarded to N.H. (Reference: SRG2021\210968). The funders had no role in study design, data collection and analysis, decision to publish or preparation of the manuscript.

## Author contributions

J.S. and N.H. conceived and designed the experiments, A.L. and N.H. developed the analytic approach, A.L. and M.F. performed the experiments, A.L. and N.H. analysed the data, A.L., L.C., R.W., S.H., and B.M. contributed materials/analysis tools, A.L. wrote the paper (original draft), and N.H. secured funding for the project. E.J. and N.H. conceptualised the interpretation and led major revisions of the manuscript. All authors (A.L., M.F., L.C., R.W., S.H., B.M., J.S., E.J., R.L., N.H.) contributed to data interpretation, critically revised the manuscript, and approved the final version.

## Competing interests

The authors declare no competing interests.
