## [Transparent Peer Review file · Communications Psychology]

Individual differences in alexithymia modulate cognition-emotion interactions in daily life ongoing experiences

Corresponding Author: Dr Nerissa Ho

Version 0:

Decision Letter:

Dear Dr Ho,

Thank you for your patience during the peer-review process. Your manuscript titled "Individual differences shape cognition-emotion interactions: Mapping real-time thought-affect relationship in daily life" has now been seen by 2 reviewers, and I include their comments at the end of this message. They find your work of interest but raised some important points. We are interested in the possibility of publishing your study in Communications Psychology, but would like to consider your responses to these concerns and assess a revised manuscript before we make a final decision on publication.

We therefore invite you to revise and resubmit your manuscript, along with a point-by-point response to the reviewers. Please highlight all changes in the manuscript text file.

Editorially, we consider that the revisions must clearly and fully address two main aspects. First, the theoretical framework needs to be improved based on the suggestions of both referees through a clearer construct definition, a better integration with the current literature, and an elaboration of rationale for the choice of alexithymia as a moderating variable. Editorially, we also invite you to integrate appraisal theories of emotion (e.g., Moors, Scherer even Lazarus), as they provide the foundation for formal conceptualization of the links between cognition and emotion. Second, we want to see the role potential confounding variables in the interpretation of the findings risen by referee 1 fully addressed, if possible with supplementary analyses. In particular, this relates to measurement confounds regarding the moderating effects of alexithymia (are the effects specific to affect, or are they more generally related to the ability to use self-report measures?), as well as to the relationship between arousal, stress, and the different thought dimensions (since they all correlate positively with arousal and stress, could this simply reflect a general tendency to report more?).

As you revise the manuscript in response to these issues, please also implement all requests in the attached Mandatory Revision Requests document. All requirements listed in this document need to be fully met, or the work will be returned to you for further revisions without peer review. This workflow is in place to increase the likelihood that the paper will be accepted for publication. It reduces the number of rounds of revision (and review) and ensures that the reviewers vet a version of the article that is compliant with journal policies. If you have any questions regarding the required revisions, please contact the journal prior to resubmission to avoid a negative outcome.

Please submit the following items:

- Revised manuscript
- Point-by-point response to the referees' comments
- Mandatory Revision Requests Table (attached).

- Cover letter (as a separate document)

via this link: Link Redacted .

** This url links to your confidential home page and associated information about manuscripts you may have submitted or are reviewing for us. If you wish to forward this email to co-authors, please delete the link to your homepage first **

Best regards,

Eva R. Pool

Eva R. Pool, PhD
Editorial Board Member
Communications Psychology
orcid.org/0000-0001-5929-1007

REVIEWER EXPERTISE:

Reviewer #1 Affective Neuroscience
Reviewer #2 Psychology of Emotion

REVIEWER REPORTS:

Reviewer #1 (Remarks to the Author):

Title: Individual differences shape cognition-emotion interactions: Mapping real-time thought-affect relationship in daily life

The current study leverages a large EMA dataset to describe thought-affect connections in daily life in young adults. Complementary analyses investigated trait moderators of these relationships, specifically alexithymia and its facets. The authors used PCA to reduce the 18 thought features rated down to 4 broad dimensions or types of thoughts, which were then examined in multilevel models with within-person affective ratings and later, between-person alexithymia scores. The study has many strengths, including 1) clearly laid aims and appropriate statistical analysis to address them, which included correction for multiple comparisons, 2) data and code are clearly presented in a publicly available repository, and 3) largely clean and intuitive figures that complement the manuscript. There are several limitations, described below, that if addressed could greatly enhance the manuscript and its impact on readers and the field. Mainly, clarification and justification concerns are noted as well as questions about the interpretation of the dimensions and patterns of results. Few of these concerns, however, render the article unsuitable for publication given the strengths mentioned above.

• Limitations

Background literature

The authors note there is little work connecting thoughts and affect in daily life; however this seems too exclusive as there are a number of areas where this type of work is ongoing, including work they cite in this article - e.g. worry/rumination/RNT literature (Hjartarson et al., 2021; Lydon-Staley et al., 2019), mind-wandering (Killingsworth & Gilbert, 2010; Welhaf et al., 2024), PTSD (Pugach et al., 2023) among others. It would be more accurate for the authors to acknowledge this ongoing work when noting the significance and novelty of their own work.

Consideration of effect of alexithymia on momentary emotion reports:

Theoretically, participants' level of access and/or awareness of the target of repeated assessment, affect, will vary across the dimension of alexithymia. Do the authors have concerns that this presents a measurement issue?

Construct definitions:

- Even brief definitions of the facets of Alexithymia would improve the manuscript given that the authors expect differential effects based upon these facets (in particular, externally oriented thinking which may be less obvious to readers who are not well-versed in this area).
- Can the authors provide more information as to why they defined arousal as how strong or intense the feeling was? Affective arousal often relates to physiological activation which is not the same as affective intensity (Russell, 1980)

Alternative interpretation of the four dimensions/types of thought:

- The authors may be oversimplifying and/or overinterpreting the dimensions of thought presented. They correctly note a limitation that they cannot “determine whether future-self-oriented thought reflects goal-directed planning, proactive coping, or prospective memory – processes that may have distinct functional outcomes”. This limitation applies to all of the dimensions and points to a broader limitation of these dimensions being described post-hoc following statistical grouping procedures.

- Relatedly, while dimension reduction in this study was useful and appropriate, there seems to be limited consideration of other possible configurations. The authors note “having identified the four dimensions of daily life thoughts...” but this breakdown is just one possible reduction configuration, not the only. Sound statistical guidelines and norms were followed and the authors do link their four dimensions to previous literature, so this is not a major concern.

- Interpretation of results: How do the authors understand the lack of specificity of thought dimensions associations with affective arousal and stress? What is it taken to mean that all four thought dimensions are positively related to arousal? For readers this raises the question of “what, if any, thought profile is associated with lower arousal?” or “Is there something missing from these four dimensions that is need to capture thoughts associated with that affective experience?” Similarly, it’s interesting that stress positively predicted 3 of the 4 thought dimensions.

- Did the authors consider any autoregressive effects or effect of time in their models given how close together some of the observations were to one another?

Minor:

- Figure 3: The authors have produced appreciably clean and clear figures of their results. Still, given the complexity of three-way interactions, it could be beneficial to have more labels/legends for each plot even if they are seemingly redundant. For example, each of the x-axis could be labeled (i.e. “Low EOT High EOT”) rather than having the single “TAS High Low” bar towards the bottom of the plot. Consider having additional labelling for the light vs bright colors in the plots. Not critical, just aiding the readers as much as possible.

- Type error: “Prior studies have found that this pattern of thought is related to the process by which people generate more concrete personal goals^{48,49} and can be important in the resolution of unhappy mood states of unhappy mood states^{11,50}”

Hjartarson, K. H., Snorrason, I., Bringmann, L. F., Ögmundsson, B. E., & Ólafsson, R. P. (2021). Do daily mood fluctuations activate ruminative thoughts as a mental habit? Results from an ecological momentary assessment study. *Behaviour Research and Therapy*, 140, 103832. <https://doi.org/10.1016/j.brat.2021.103832>

Killingsworth, M. A., & Gilbert, D. T. (2010). A wandering mind is an unhappy mind. *Science (New York, N.Y.)*, 330(6006), 932. <https://doi.org/10.1126/science.1192439>

Lydon-Staley, D. M., Kuehner, C., Zamoscik, V., Huffziger, S., Kirsch, P., & Bassett, D. S. (2019). Repetitive negative thinking in daily life and functional connectivity among default mode, fronto-parietal, and salience networks. *Translational Psychiatry*, 9(1), 1–12. <https://doi.org/10.1038/s41398-019-0560-0>

Pugach, C. P., May, C. L., & Wisco, B. E. (2023). Positive emotion in posttraumatic stress disorder: A global or context-specific problem? *Journal of Traumatic Stress*, 36(2), 444–456. <https://doi.org/10.1002/jts.22928>

Russell, J. A. (1980). A circumplex model of affect. *Journal of Personality and Social Psychology*, 39(6), 1161–1178. <https://doi.org/10.1037/h0077714>

Webb, C. A., Tierney, A. O., Brown, H. A., Forbes, E. E., Pizzagalli, D. A., & Ren, B. (2022). Spontaneous Thought Characteristics are Differentially Related to Heightened Negative Affect vs. Blunted Positive Affect in Adolescents: An Experience Sampling Study. *JCPP Advances*, 2(4), e12110. <https://doi.org/10.1002/jcv2.12110>

Welhaf, M. S., Mata, J., Jaeggi, S. M., Buschkuhl, M., Jonides, J., Gotlib, I. H., & Thompson, R. J. (2024). Mind-wandering in daily life in depressed individuals: An experience sampling study. *Journal of Affective Disorders*, 366, 244–253. <https://doi.org/10.1016/j.jad.2024.08.111>

Reviewer #2 (Remarks to the Author):

The work presented by the authors shows that examining people’s ongoing thoughts in daily life provides a valuable way to reveal how cognition and emotion are linked in context-dependent ways. This work can contribute to the development of tailored interventions aimed at improving maladaptive patterns of thinking and feeling. The work uses multidimensional experience sampling as a method to map thought-affect interactions in daily life. The language is clear and the manuscript is well-written and easy to follow. Nevertheless, the manuscript would strongly benefit from some revisions upon addressing the comments reported below:

- Introduction, second paragraph:

The authors should provide stronger theoretical arguments for the selection of their key dimensions. If the premise is that thoughts have a direct impact on affect, a framework involving concepts such as rumination should be included somewhere.

This would be consistent with the abundant literature on mind-wandering that the authors are citing.

- Introduction, third paragraph:

Can the authors provide deeper arguments for choosing alexithymia as a stable personality trait of interest? Why not neuroticism, for example? It also shapes thought-affect interactions and has a stronger influence in neuropsychiatric disorders. Besides this, it has widely studied brain-behavior links (See Tseng und Poppenk, 2020 for a recent approach), in case the authors want to argue on the neurobiological bases of the trait. This is only an example, but it calls out for a more well-grounded theory on the selection of alexithymia.

Figure 1 depicts the correlation between TAS-20 and DASS scores. However, it is the first time that the DASS scale appears in the manuscript. Can the authors shortly introduce the rationale behind this inclusion in the introduction?

Results:

The authors described "subjective stress" in the introduction. Was this the construct analysed in the study, or was it just "stress"? If it was the first case, it would explain why stress did not predict sensory engagement. This can give a hint on the involvement of interoceptive disruptive signals during the experience sampling (see Suksasilp & Garfinkel, 2022). Can the authors comment or clarify this?

Echoing the comment from above, the authors also showed that introspective people were more task focused when feeling stressful. This finding should also be considered in the discussion.

Methods:

One criticism of the TAS scale is that people with alexithymia will not provide accurate responses due to the difficulties mapping their feelings. Did the authors identified ceiling effects in this scale or controlled for potential outliers?

Typically, experience sampling experiments analyse nested observations within individuals; therefore, hierarchical or multilevel models are the choice. Why did the authors choose the run linear mixed effects models? Even when they are mathematically the same, LMMs and MLM focus on different outputs. Can the authors provide an explanation about the selection of their statistical model?

Minor comments:

- Abstract:

Authors should include a short statement about the sample. Were they students, age range, N=??

- While it is a well-written draft, it will benefit from correction in the citation style. It is true that several studies already provided the bases for the current investigation, but the authors do not need to cite them all!. Some citations are misplaced or do not support the concepts or methods that claim. Besides this, they are too many. I suggest the authors to focus on the seminal and on the most recent or important articles.

Communications Psychology is committed to improving transparency in authorship. As part of our efforts in this direction, we are now requesting that all authors identified as 'corresponding author' create and link their Open Researcher and Contributor Identifier (ORCID) with their account on the Manuscript Tracking System prior to acceptance. ORCID helps the scientific community achieve unambiguous attribution of all scholarly contributions. You can create and link your ORCID from the home page of the Manuscript Tracking System by clicking on 'Modify my Springer Nature account' and following the instructions in the link below. Please also inform all co-authors that they can add their ORCID to their accounts and that they must do so prior to acceptance.

Version 1:

Decision Letter:

Dear Dr Ho,

Your manuscript titled "Individual differences in alexithymia shape cognition-emotion interactions in daily life ongoing experiences" has now been seen by our reviewers, whose comments appear below. In light of their advice I am delighted to say that we are happy, in principle, to publish a suitably revised version in Communications Psychology.

We therefore invite you to revise your paper one last time to address the remaining concerns of our reviewers and a list of editorial requests. At the same time we ask that you edit your manuscript to comply with our format requirements and to maximise the accessibility and therefore the impact of your work.

EDITORIAL REQUESTS:

SUBMISSION INFORMATION:

OPEN ACCESS:

*** TRANSPARENT PEER REVIEW:** Communications Psychology uses a transparent peer review system. On author request, confidential information and data can be removed from the published reviewer reports and rebuttal letters prior to publication. If you are concerned about the release of confidential data, please let us know specifically what information you would like to have removed. Please note that we cannot incorporate redactions for any other reasons.

*** CODE AVAILABILITY:** All Communications Psychology manuscripts must include a section titled "Code Availability" at the end of the methods section. We require that the custom analysis code supporting your conclusions is made available in a publicly accessible repository at this stage; please choose a repository that generates a digital object identifier (DOI) for the code; the link to the repository and the DOI must be included in the Code Availability statement. Publication as Supplementary Information will not suffice.

*** DATA AVAILABILITY:**

Link Redacted

Best regards,

Jennifer Bellingtier

Jennifer Bellingtier, PhD
Senior Editor
Communications Psychology

Eva R. Pool, PhD
Editorial Board Member
Communications Psychology
orcid.org/0000-0001-5929-1007

REVIEWER EXPERTISE:

Reviewer #1 Affective Neuroscience
Reviewer #2 Psychology of Emotion

REVIEWERS' COMMENTS:

Reviewer #1 (Remarks to the Author):

The authors have made thoughtful and thorough revisions based on the reviewer feedback. Additional background information and clarification throughout the manuscript improve the readability and impact of the paper. I am largely satisfied with the revisions made.

The only point that I have lingering questions about is possible reporting differences associated with alexithymia. In response to my previous point ("Theoretically, participants' level of access and/or awareness of the target of repeated assessment, affect, will vary across the dimension of alexithymia. Do the authors have concerns that this presents a measurement issue?"), the authors performed analyses to rule out "significant differences in reported valence, arousal, and stress across all alexithymia measures". However, I still wonder about aspects of reporting besides response magnitude, such as number of responses, time to respond/time per item or per survey, and also within-person variability in responses on items.

These features are commonly examined in EMA research to identify 'careless responding' (Jaso, B. A., Kraus, N. I., & Heller, A. S. (2022). Identification of careless responding in ecological momentary assessment research: From posthoc analyses to real-time data monitoring. *Psychological Methods*, 27(6), 958–981. <https://doi.org/10.1037/met0000312>), and the authors indeed used a number of responses and time to response as exclusion criteria. Although there do not appear to be citations for the chosen criteria, which would be helpful. It would also be helpful to see descriptive statistics of these response features for included individuals and observations. In part because these response features can also help us understand - how- people are completing the surveys, beyond what they report.

My lingering questions are whether these response features are associated with alexithymia. Are individuals who score higher on alexithymia submitting fewer surveys? responding faster or slower (perhaps indicating degree of deliberation)? or reporting within a more narrow range (e.g. only using 10 pts of a 100 pt scale across all observations)? If so, we should consider if/how that impacts these results and broader affect EMA research. If not, why? If not, doesn't that undermine the definition of alexithymia as a difficulty accessing and identifying feeling states?

These seem like important questions to me, but I also do not feel like the manuscript is unsuitable for publication without further investigation. Moreover, alexithymia is not my expertise, thus, I will leave it to the editor's discretion on whether more revision or investigation is necessary. Thank you for the opportunity to review this promising work.

Reviewer #2 (Remarks to the Author):

The authors have addressed all the questions/concerns raised by the reviewer. I do not have further comments.

Reviewer 1

Overall comment:

The current study leverages a large EMA dataset to describe thought-affect connections in daily life in young adults. Complementary analyses investigated trait moderators of these relationships, specifically alexithymia and its facets. The authors used PCA to reduce the 18 thought features rated down to 4 broad dimensions or types of thoughts, which were then examined in multilevel models with within-person affective ratings and later, between-person alexithymia scores. The study has many strengths, including 1) clearly laid aims and appropriate statistical analysis to address them, which included correction for multiple comparisons, 2) data and code are clearly presented in a publicly available repository, and 3) largely clean and intuitive figures that complement the manuscript. There are several limitations, described below, that if addressed could greatly enhance the manuscript and its impact on readers and the field. Mainly, clarification and justification concerns are noted as well as questions about the interpretation of the dimensions and patterns of results. Few of these concerns, however, render the article unsuitable for publication given the strengths mentioned above.

Main issues	Exact comments/questions	Author Reply
R1.1 Background literature	The authors note there is little work connecting thoughts and affect in daily life; however this seems too exclusive as there are a number of areas where this type of work is ongoing, including work they cite in this article - e.g. worry/rumination/RNT literature (Hjartarson et al., 2021; Lydon-Staley et al., 2019), mind-wandering (Killingsworth & Gilbert, 2010; Welhaf et al., 2024), PTSD (Pugach et al., 2023) among others. It would be more accurate for the authors to acknowledge this ongoing work when noting the significance and novelty of their own work.	We agree with the reviewer that our initial phrasing did not sufficiently reflect the previous and on-going work on thought-affect relationships. In the revised introduction, we now explicitly acknowledged contribution from several relevant literatures, as well as incorporated a stronger theoretical foundation by referring to cognitive approaches to emotion, particularly cognitive appraisal theories. We also emphasized the significance/novelty of our own work lies in: (1) we examined a wider range of thought and affect dimensions simultaneously using a real-time assessment, and (2) demonstrated that such thought-affect relationships are not uniform across individuals but depend on emotion-relevant traits (alexithymia and its sub-facets). We have revised the first part of the Introduction: “Thought patterns and emotional experience are intimately intertwined. In daily life, a substantial proportion of self-generated thought carries an affective tone¹. Maladaptive thought patterns are common in emotional disorders such as depression, anxiety, and post-traumatic stress^{2,3,4}, with strong evidence that repetitive negative thinking (rumination and worry) sustains and exacerbates negative mood^{5,6}. Therapeutic approaches such as cognitive-behavioural and mindfulness-based interventions target

		these thought patterns to improve emotional well-being, highlighting the functional importance of thought-affect interactions^{7,8}. Beyond clinical constructs, cognitive models of emotion, particularly appraisal theories, also provide a broader foundation for understanding thought-affect interactions^{9,10}. These theories propose that cognitive evaluations of external and internal events along different dimensions (e.g., goal congruence, importance, agency, intentionality, coping potential) predict the quality of emotional experience and core affective dimensions including valence and arousal^{11,12}. Complementing these accounts, recent research on self-generated thought shows that negative affect, particularly sad mood, is consistently associated with intrusive, past-focused mind wandering^{13,14,15}, whereas positive affect tends to accompany task-related, prospective and socially oriented thought patterns^{16,17}. This thought-affect relationship has been formalized in the content regulation hypothesis^{18,19}, which suggests that the affective outcomes of task-unrelated thoughts are not always negative, but depend on their experiential contents. Despite this extensive investigation of cognition-emotion interactions, much existing work relies on controlled laboratory tasks or retrospective self-reports that are prone to memory biases. To improve ecological validity, real-life experience sampling methods are increasingly used to characterize the relationship between thought patterns and affective valence^{17,20,21}. However, most of these studies focused on specific sets of thought (e.g., mind wandering, rumination), limiting a comprehensive understanding of how heterogeneous thought patterns map onto broader affective experiences in daily life. Moreover, beyond valence, high arousal has been linked to increased self-related contemplation and task-focus, whereas low arousal is associated with inattentive or off-task states^{22,23}. Subjective stress (i.e., personal perception of stress), though less frequently studied, also shapes cognition, influencing thought patterns depending on context and potentially modulating adaptive or maladaptive coping^{24,25}. These findings collectively motivate the inclusion of valence, arousal, and stress as key affective dimensions in our study.”
--	--	---

R1.2 Consideration of effect of alexithymia on momentary emotion reports:	Theoretically, participants' level of access and/or awareness of the target of repeated assessment, affect, will vary across the dimension of alexithymia. Do the authors have concerns that this presents a measurement issue?	We thank the reviewer for drawing our attention to this potential confounding factor. We agree that at a theoretical level, high alexithymia is linked to reduced access/awareness of their mental states, which might render their introspective reports less accurate. While we could not completely rule out the possibilities that variation in the level of access/awareness of affect across participants might cause measurement issue with the repeated affect reports, we conducted additional analyses to examine whether alexithymia measures moderate affective ratings. This is reported in the Results section and in the Supplementary Materials: “Given the potential concern that people with higher alexithymia may have more limited access to their affective states, we also conducted analyses to examine whether alexithymia measures moderate affective ratings. Using total alexithymia scores and all three subscales to predict each momentary affect rating while controlling for age, gender, and DASS-total scores, we found no significant differences in reported valence, arousal, and stress across all alexithymia measures (all $p > 0.05$; see Supplementary S12-13), except for a weak linkage of higher DDF to lower arousal ($\beta = -0.24$, 95%CI [-0.46, -0.02], $p = 0.034$) and lower stress ($\beta = -0.25$, 95%CI [-0.44, -0.05], $p = 0.015$). Hence, we consider that these results provide no strong evidence that alexithymia meaningfully biases momentary affect reports in our study.” In fact, employing ESM for measuring thoughts and emotions would help to supplement such potential inaccuracy/bias associated with retrospective self-report. This method has been successfully applied in clinical populations in which reporting biases are common (e.g., depression; Myin-Germeys et al., 2018). Additionally, we note that there is also evidence showing high alexithymia traits are at times linked to better access to their mental states such as stress (e.g., Ellison et al., 2021) compared to people with low alexithymia. Taken together, we consider that these results provide no strong evidence that alexithymia meaningfully biases momentary affect reports in our study. Nonetheless, we highlight in the revised manuscript that future research would benefit from incorporating more objective measures of affect to complement subjective self-reports.
--	--	--

		“Several limitations warrant caution. Our measures of affective states might capture a mixture of chronic moods and transient episodes; future studies could disambiguate these temporal scales. We also note that although our PCA-driven thought patterns can identify higher-level structural dimensions of thoughts, we cannot determine the specific cognitive processes that underlie each thought pattern nor the exact purpose or function of a given thought episode. Therefore, these thought patterns should be interpreted with caution. Future studies could enrich these interpretations, for example, by directly assessing the purpose of ongoing thoughts (e.g., goal-directed planning, emotion regulation, fantasy), or by combining experience sampling with think-aloud approaches⁷² to measure more fine-grained content within each thought dimension. Additionally, future studies could benefit from asking participants to report on thoughts that are focused on internal bodily senses to better understand how interoceptive attention relates to affect, which may be distinct from that of visual or auditory senses⁵⁹, and thus, might be moderated by alexithymia in a different manner. Objective measures of affect (e.g., physiological methods) would ideally also be incorporated to complement subjective self-reports. Finally, our predominantly female, Western undergraduate sample limits generalizability; replication in more diverse and clinical populations will be crucial.”
R1.3 Construct definitions	Even brief definitions of the facets of Alexithymia would improve the manuscript given that the authors expect differential effects based upon these facets (in particular, externally oriented thinking which may be less obvious to readers who are not well-versed in this area).	We appreciate the reviewer’s note concerning this missing detail. We have included more descriptions of these facets in paragraph 3: “In the current study, we adopt alexithymia²⁶, a multi-faceted personality trait reflecting atypical emotional awareness, as the construct for examining individual differences in thought-affect interactions. This is theoretically relevant because alexithymia is composed of three sub-facets that are conceptually defined by different disruptions in the integration of cognitive and emotional processes²⁷: difficulty identifying feelings (DIF), difficulty describing feelings (DDF), and externally oriented thinking (EOT). The DIF facet captures the degree of difficulty in recognizing one’s feelings and distinguishing them from physical sensations, the DDF facet reflects difficulty in verbalizing one’s feelings to other people, and the EOT facet describes the tendency to focus on external events rather than internal feelings²⁸. Based on the attention–appraisal model of alexithymia²⁹, people with high alexithymia have reduced attention to emotional cues (high EOT) and/or impaired cognitive appraisal (high DIF/DDF).

		The three-process model of emotional awareness^{30,31} also proposed that alexithymia involves disruptions in accessing and communicating cognitive meanings of emotions (high DIF/DDF), detecting affective signals for setting cognitive priority (high EOT), and applying cognitive control over emotional states through attention and working memory (all three sub-facets). Yet, how these facets shape momentary thought–affect relationships remains largely unexplored in naturalistic, daily life contexts.”
	Can the authors provide more information as to why they defined arousal as how strong or intense the feeling was? Affective arousal often relates to physiological activation which is not the same as affective intensity (Russell, 1980).	Thanks for raising this important conceptual distinction. We agree that in the circumplex model (Russell, 1980), affective arousal refers to physiological activation (e.g., feeling activated, energized, tense), which is conceptually distinct from affective intensity. At the same time, other approaches either blend the two constructs (e.g., Clore, Ortony, & Foss, 1987; Lang et al., 1994) or use strength as an accessible index of overall arousal in self-reports (e.g., Citron et al., 2014). Further, work on affect intensity (e.g., Larsen & Diener, 1987) highlights that measures of emotional “strength” typically integrate both subjective intensity and perceived bodily activation, particularly in daily-life settings where people do not easily separate these components. Therefore, we consider our wording (“How were you feeling? My feeling was very strong (0 = Not at all; 10 = Completely)”) largely captures the aspect of arousal that is reliably accessible to participants in everyday contexts. This wording was also chosen to avoid potential confusion regarding jargon terms such as “arousal” or “activation”. In the Methods section where we first introduced affective ratings, we added a sentence to explain that responses to this “arousal” item may reflect a mixture of perceived emotional strength and bodily activation. “Affective states were rated by valence (“I was very happy”, 0 = very sad, 10 = very happy), arousal (“My feeling was very strong”, 0 = Not at all, 10 = Completely) and stress (“I was very stressed”, 0 = Not at all, 10 = Completely) in response to the question prompt “How were you feeling?”. It should be noted that the wording of the arousal statement in terms of “strong” was chosen to avoid potential confusion with jargon (e.g., “activated”, “aroused”), and

		responses may reflect a mixture of perceived emotional strength and bodily activation. Distribution of ratings on affective states of valence, arousal, and stress is presented as histograms in Fig. 1C.”
R1.4 Alternative interpretation of the four dimensions/types of thought:	The authors may be oversimplifying and/or overinterpreting the dimensions of thought presented. They correctly note a limitation that they cannot “determine whether future-self-oriented thought reflects goal-directed planning, proactive coping, or prospective memory – processes that may have distinct functional outcomes”. This limitation applies to all of the dimensions and points to a broader limitation of these dimensions being described post-hoc following statistical grouping procedures. Relatedly, while dimension reduction in this study was useful and appropriate, there seems to be limited consideration of other possible configurations. The authors note “having identified the four dimensions of daily life thoughts...” but this breakdown is just one possible reduction configuration, not the only. Sound statistical guidelines and norms were followed and the authors do link their four dimensions to previous literature, so this is not a major concern.	We are grateful to the reviewer’s constructive reminder about this concern. We completely agree that our measures cannot determine the specific cognitive processes that underlie each thought pattern (e.g., planning vs. problem-solving vs. fantasizing), nor can they capture the purpose or function of a given thought episode. We have now clarified that this limitation applies to all of the dimensions identified in our study, not only future-self-oriented thought, and added caution against overinterpretation. We also highlighted in the Methods section that our goal for applying PCA was to identify higher-level structural dimensions of ongoing cognition using data-driven methods. While there might be different ways to approach dimension reduction, by using the same methods as previous studies, we can compare our results with existing studies and reaffirm that these thought structures do have emerged repeatedly across prior work (e.g., Mulholland et al., 2023; Chitiz et al., 2025), supporting the postulation that these thought dimensions have stable neural correlates across different settings (e.g., Smallwood et al., 2021). In the revised manuscript, we now also highlight opportunities for future research to enrich these interpretations - for example, by directly assessing the purpose of ongoing thoughts (e.g., planning, emotion regulation), or by combining experience sampling with think-aloud approaches, which might capture more fine-grained content/purpose within each thought pattern. Please see the revised text in the above response to R1.2 Consideration of effect of alexithymia on momentary emotion reports.
R1.5 Lack of specific relationships	How do the authors understand the lack of specificity of thought dimensions associated with affective arousal and stress? What is it taken to mean that all four thought dimensions are positively related to arousal? For readers this raises the question of “what, if any, thought profile is	We thank the reviewer for drawing our attention to this confusion. For clarification, we elaborated in the Methods section that labelling of these thought dimensions is based on the

between thought and arousal/stress	associated with lower arousal?” or “Is there something missing from these four dimensions that is need to capture thoughts associated with that affective experience?” Similarly, it’s interesting that stress positively predicted 3 of the 4 thought dimensions.	high loading variables of the PCA components from both positive and negative directions (negative variables are attached with the opposite meaning). “Thought components were labelled based on the most highly loaded features from both positive and negative directions, with negative variables interpreted as representing the opposite meanings of the corresponding features.” Thus, thought profiles associated with low arousal/stress correspond to the opposite ends of these thought dimensions such as “pleasant deliberate” thoughts (the opposite pole of “intrusive distraction”) and “off-task internally focused” thoughts (the opposite pole of external “task-focus”). This result is consistent with prior research showing low arousal is linked with decreased task-relevant focus (e.g., Kiss & Linnell, 2024; Sakuragi et al., 2024). We have made this interpretation explicit in the Results section. “Arousal positively predicted all dimensions of thought (all $p < 0.001$), meaning that higher arousal was associated with increased engagement across all types of thought, while lower arousal would correspond to the opposite ends of these thought dimensions such as “pleasant deliberate” thoughts (the opposite pole of “intrusive distraction”) and “off-task internally focused” thoughts (the opposite pole of “task-focus”). Association between lower arousal and decreased task-relevant focus is consistent with prior research^{22,23}. Similarly, stress positively predicted all dimensions of thought (all $p < 0.001$), except for sensory engagement ($\beta = -0.002$, 95% CI [-0.03, 0.03], $p = 0.873$).” Moreover, although arousal showed positive associations with all four thought dimensions, two of these relationships were not simple main effects but involved valence and arousal interactions. In particular, “intrusive distraction” and “on-task” dimensions increased during negative high-arousal states and positive high-arousal states, respectively. This demonstrates that the effects of arousal were not uniform across thought dimensions but depended on the valenced meaning of the arousal. We have highlighted this in the Results section as well. “This demonstrates that the effects of arousal were not uniform across these thought dimensions but depended on the emotional valence of the affective states. No significant
---	---	---

		interaction effects were observed for future-self focus ($F(1,5047.47) = 0.79, p = 0.376$) and for sensory engagement ($F(1,4993.456) = 1.44, p = 0.231$).”
R1.6 Effect of time	Did the authors consider any autoregressive effects or effect of time in their models given how close together some of the observations were to one another?	We appreciate the reviewer’s consideration. We did not include the effect of time or autoregressive effects in our original models, as these were not of primary interest. We have re-run the analyses including “time” (probe position with each day) as a control variable. The significance of all the main findings remains unchanged. The results of these additional analyses are reported in the Results section and in the updated Supplementary Materials. “Finally, to address the possibility that thoughts and affects might be confounded by the time of sampling, we reran the analyses including the probe order (within each day) as an additional covariate in the LMMs. All key significant effects remained unchanged (see Supplementary S14–S18).”
Minor issues	Exact comments/questions	Author Reply
R1.7 Figure 3 label	The authors have produced appreciably clean and clear figures of their results. Still, given the complexity of three-way interactions, it could be beneficial to have more labels/legends for each plot even if they are seemingly redundant. For example, each of the x-axis could be labeled (i.e. “Low EOT High EOT”) rather than having the single “TAS High Low” bar towards the bottom of the plot. Consider having additional labelling for the light vs bright colors in the plots. Not critical, just aiding the readers as much as possible.	We agree with the reviewer’s suggestion. We have added additional x-axis labels to Figure 3.
R1.6 Type error:	“Prior studies have found that this pattern of thought is related to the process by which people generate more concrete personal goals ^{48,49} and can be important in the resolution of unhappy mood states of unhappy mood states ^{11,50} ”	Thanks for noticing - we have corrected this.

Reviewer 2

Overall comment:

The work presented by the authors shows that examining people's ongoing thoughts in daily life provides a valuable way to reveal how cognition and emotion are linked in context-dependent ways. This work can contribute to the development of tailored interventions aimed at improving maladaptive patterns of thinking and feeling. The work uses multidimensional experience sampling as a method to map thought-affect interactions in daily life. The language is clear and the manuscript is well-written and easy to follow. Nevertheless, the manuscript would strongly benefit from some revisions upon addressing the comments reported below:

Main issues	Exact comments/questions	Author Reply
R2.1 Introduction, second paragraph:	The authors should provide stronger theoretical arguments for the selection of key dimensions. If the premise is that thoughts have a direct impact on affect, a framework involving concepts such as rumination should be included somewhere. This would be consistent with the abundant literature on mind-wandering that the authors are citing.	We appreciate the reviewer pointing out that the Introduction could benefit from a stronger theoretical foundation. In our revision, we have incorporated more frameworks (e.g., cognitive appraisal) to provide a broad theoretical context for studying thought-emotion interactions. We also recognize the substantial contribution of rumination/repetitive negative thinking literature in establishing the thought-mood connections in psychopathology, so we have now explicitly mentioned it in the Introduction (Please see our response to Reviewer 1 above R1.1 Background literature). Moreover, we would like to clarify that our study does not assume a direct causal impact of thoughts on affect nor specifically utilizes rumination/repetitive negative thinking as a framework for understanding thought-affect connection. Rather, our aim was to examine how multiple, heterogenous dimensions of ongoing thought covary with affective states in daily life. These dimensions of thought are derived from a broad range of thought features using PCA. Rumination would likely be the type of thoughts with high loadings on negative emotion, past-oriented, self-focused, intrusive and distracting, that is, close to the second thought dimension (“intrusive distraction”) in our study. We have made this link between the “intrusive distraction” component and the rumination literature more explicit in our Discussion.

		“Intrusive distraction, in turn, tracked stress and sadness, particularly under high arousal and in solitude, corroborating prior work linking repetitive negative thinking to negative mood and increased symptoms of depression^{5,52,57}.”
R2.2 Introduction, third paragraph:	Can the authors provide deeper arguments for choosing alexithymia as a stable personality trait of interest? Why not neuroticism, for example? It also shapes thought-affect interactions and has a stronger influence in neuropsychiatric disorders. Besides this, it has widely studied brain-behavior links (See Tseng und Poppenk, 2020 for a recent approach), in case the authors want to argue on the neurobiological bases of the trait. This is only an example, but it calls out for a more well-grounded theory on the selection of alexithymia.	Thanks for highlighting the importance of clarifying the theoretically grounded rationale for our selection of alexithymia as the key personality trait. In the revised Introduction, we now more explicitly outline why alexithymia is especially well suited for understanding cognition-emotion interactions. Please see the revised text in response to Reviewer 1 above to R1.3 Construct definitions Although traits such as neuroticism also influence affect, we consider that alexithymia more specifically targets the cognition-emotion/affect interactions, making it a more theoretically relevant choice.
R2.3 Figure 1’s DASS-score	Figure 1 depicts the correlation between TAS-20 and DASS scores. However, it is the first time that the DASS scale appears in the manuscript. Can the authors shortly introduce the rationale behind this inclusion in the introduction?	We thank the reviewer for drawing our attention to this oversight. We have introduced the rationale for DASS towards the end of the Introduction: “Further, as general psychological distress can influence both affect and thought independently of alexithymia, we controlled for baseline distress using the 21-item Depression, Anxiety, and Stress Scale (DASS-21)⁴⁰. This enables us to isolate the contribution of alexithymia traits from distress-related factors that may also moderate thought-affect relationships.”
R2.4 Results regarding stress and sensory engagement	The authors described “subjective stress” in the introduction. Was this the construct analysed in the study, or was it just “stress”? If it was the first case, it would explain why stress did not predict sensory engagement. This can give a hint on the involvement of interoceptive disruptive signals during the experience sampling (see Suksasilp & Garfinkel, 2022). Can the authors comment or clarify this? Echoing the comment from above, the authors also showed that	Thank you for raising this point regarding the distinction between “subjective stress” and “stress”, as the former refers to the subjective/perceived nature of feeling stressed and the latter can broadly include subjective perceptions of stress, physiological stress responses, and objectively presented stressors in the environment. In our study, we assessed “subjective stress” using the rating item “I was very stressed, 0 = Not at all , 10 = Completely”. We have made the distinction more explicit in Introduction where we introduced this term:

introspective people were more task focused when feeling stressful. This finding should also be considered in the discussion.

“Subjective stress (i.e., personal perception of stress), though less frequently studied, also shapes cognition, influencing thought patterns depending on context and potentially modulating adaptive or maladaptive coping^{24,25}.”

If we understood the reviewer’s suggestion correctly, the use of subjective stress may partly explain why stress did not predict sensory engagement because subjective stress ratings (unlike physiological stress responses) may not reliably produce changes in internal sensory or interoceptive processing (Suksasilp & Garfinkel, 2022). We note, however, that the “sensory engagement” in our study refers to the dimension related to the modality or form of thoughts (highly loaded on “Images”, “Sounds”, “Words”) and linked to more socially oriented thought content (“People”) and positive “Emotion”, and thus, were also reported more often in social rather than solitary context. Indeed, prior work studying ongoing thought also similarly showed that this thought pattern was associated with increased activation in visual and auditory systems during movie-watching (Wallace et al., 2025), reflecting engagement in leisure activities with high multi-sensory focus on the modality of thoughts (e.g., TV, music; Mulholland et al., 2025).

We have revised the Discussion to interpret why stress did not predict sensory engagement:

“Finally, sensory engagement, representing the emphasis on the modality of thoughts (e.g., sounds, images), was associated with happiness, arousal, and being in a social setting, consistent with research showing that multisensory stimulation and auditory attention amplify positive affect^{58,59}. Interestingly, sensory engagement was the only thought dimension not predicted by stress, despite some prior work linking heightened sensory processing to increased stress⁶⁰. One explanation is that this kind of sensory engagement captured in our study differs qualitatively from the sensitivity to external and internal senses typically associated with stress, especially heightened interoception⁶¹.”

We also added a sentence in the Limitation noting that future studies could better disentangle attention focusing on external senses and internal senses:

		“Additionally, future studies could benefit from asking participants to report on thoughts that are focused on internal bodily senses to better understand how interoceptive attention relates to affect, which may be distinct from that of visual or auditory senses⁵⁹, and thus, might be moderated by alexithymia in a different manner. Objective measures of affect (e.g., physiological methods) would ideally also be incorporated to complement subjective self-reports.” Regarding the findings on people with higher scores on EOT/externally oriented thinking (“less introspective”) on the TAS-20 scale, we hesitate to assume these individuals have reduced interoceptive ability or less interoceptive attention, as there is evidence showing that the EOT facet is not really associated with interoception, despite the connection between overall alexithymia and altered interoception (Gaggero et al., 2022). To avoid potential over-interpretation, our interpretation for EOT in this study would, therefore, mainly focus on its relevance as a marker of focusing attention on external environments over internal mental states.
R2.5 Methods: concern about accuracy in report	One criticism of the TAS scale is that people with alexithymia will not provide accurate responses due to the difficulties mapping their feelings. Did the authors identified ceiling effects in this scale or controlled for potential outliers?	We examined the distributions of the TAS scales and found no ceiling effects in the total scale and its subscales. Across all scales, the percentage of maximum or near-maximum scores was well below standard thresholds (15%) for defining ceiling effects (Terwee et al., 2007), and the distribution was approximately symmetric (skewness close to 0). A more detailed breakdown is shown below: For TAS-total, the highest score observed was 80, with 0.53% of participants scoring at this value and 2.11% scoring within five points of it, skewness was -0.046. For TAS-DIF, the maximum score was 35, with 0.53% of participants at the maximum and 3.16% within five points; skewness was -0.008. For TAS-DDF, the maximum score was 35, with 0.53% scoring the maximum and 4.21% within five points; skewness was 0.031.

		For TAS-EOT, the maximum observed value was 37, with 0.53% of participants scoring at this value and 0.53% within five points; skewness was 0.39.
R2.6 Rationale for using Linear Mixed Models	Typically, experience sampling experiments analyse nested observations within individuals; therefore, hierarchical or multilevel models are the choice. Why did the authors choose the run linear mixed effects models? Even when they are mathematically the same, LMMs and MLM focus on different outputs. Can the authors provide an explanation about the selection of their statistical model?	Thank you for raising this point. We would like to note that the LMM specification we used (random intercepts for participants) fully captures the multilevel/nested structure of our ESM data. We agree that MLM and LMM frameworks focus on different outputs (for MLM, variance partitioning and intraclass correlations, and for LMM, fixed-effect estimates and model fit). Our analytic aim was to estimate fixed effects - specifically, within-person associations between affect and thought, between-person effects of alexithymia (TAS-20), and cross-level interactions where alexithymia moderates thought-affect links. For these purposes, LMMs provide the same hierarchical structure as MLMs while aligning with the modelling methods used in prior experience-sampling research employing similar sampling paradigms of thoughts (e.g., McKeown et al., 2021; Mulholland et al., 2025; Myin-Germeys et al., 2018). We have clarified this rationale in the Methods section by adding:

		Thank you for raising this point. We would like to note that the LMM specification we used (random intercepts for participants) fully captures the multilevel/nested structure of our ESM data. We agree that MLM and LMM frameworks focus on different outputs (for MLM, variance partitioning and intraclass correlations, and for LMM, fixed-effect estimates and model fit). Our analytic aim was to estimate fixed effects - specifically, within-person associations between affect and thought, between-person effects of alexithymia (TAS-20), and cross-level interactions where alexithymia moderates affect–thought links. For these purposes, LMMs provide the same hierarchical structure as MLMs while aligning with the modelling conventions used in prior experience-sampling research employing similar sampling paradigms (e.g., McKeown et al., 2021; Mulholland et al., 2025; Myin-Germeys et al., 2018). We have clarified this rationale in the Methods section by adding: “We analyzed the experience-sampling data (PCA scores) using LMMs with random intercepts for participants to account for the nested structure of observations within individuals⁴⁶.”
Minor issues	Exact comments/questions	Author Reply
R2.7 Abstract	Authors should include a short statement about the sample. Were they students, age range, N=??	We have updated the abstract to address this point.
R2.8 Citation style	While it is a well-written draft, it will benefit from correction in the citation style. It is true that several studies already provided the bases for the current investigation, but the authors do not need to cite them all!. Some citations are misplaced or do not support the concepts or methods that claim. Besides this, they are too many. I suggest the authors to focus on the seminal and on the most recent or important articles.	We agree that the citation style could be improved. We have removed redundant citations (cut down ~10 citations) and made sure no citations are misplaced.

Citations

- Chitiz, L., Mckeown, B., Mulholland, B., Wallace, R., Goodall-Halliwell, I., Ping-Ho, N. S., Konu, D., Poerio, G. L., Wammes, J., Milham, M., Klein, A., Jefferies, E., Leech, R., & Smallwood, J. (2025). Mapping cognition across lab and daily life using Experience-Sampling. *Consciousness and Cognition*, *131*, 103853. <https://doi.org/10.1016/j.concog.2025.103853>
- Citron, F. M. M., Gray, M. A., Critchley, H. D., Weekes, B. S., & Ferstl, E. C. (2014). Emotional valence and arousal affect reading in an interactive way: Neuroimaging evidence for an approach-withdrawal framework. *Neuropsychologia*, *56*, 79–89. <https://doi.org/10.1016/j.neuropsychologia.2014.01.002>
- Clore, G. L., Ortony, A., & Foss, M. A. (1987). The psychological foundations of the affective lexicon. *Journal of Personality and Social Psychology*, *53*(4), 751–766. <https://doi.org/10.1037/0022-3514.53.4.751>
- Ellison, W. D., Trahan, A. C., Pinzon, J. C., Gillespie, M. E., Simmons, L. M., & King, K. Y. (2020). For whom, and for what, is experience sampling more accurate than retrospective report? *Personality and Individual Differences*, *163*, 110071. <https://doi.org/10.1016/j.paid.2020.110071>
- Gaggero, G., Dellantonio, S., Pastore, L., Sng, K. H. L., & Esposito, G. (2022). Shared and unique interoceptive deficits in high alexithymia and neuroticism. *PLOS ONE*, *17*(8), e0273922. <https://doi.org/10.1371/journal.pone.0273922>
- Kiss, L., & Linnell, K. J. (2024). The role of mood and arousal in the effect of background music on attentional state and performance during a sustained attention task. *Scientific Reports*, *14*(1), 9485. <https://doi.org/10.1038/s41598-024-60218-z>
- Larsen, R. J., & Diener, E. (1987). Affect intensity as an individual difference characteristic: A review. *Journal of Research in Personality*, *21*(1), 1–39. [https://doi.org/10.1016/0092-6566\(87\)90023-7](https://doi.org/10.1016/0092-6566(87)90023-7)
- Lang, P. J. (1994). The motivational organization of emotion: Affect reflex connections. In S. H. M. van Goozen & N. E. Van de Poll (Eds.), *Emotions: Essays on emotion theory* (pp. 61–93). Hillsdale, NJ: Erlbaum.
- Luminet, O., Nielson, K. A., & Ridout, N. (2021a). Having no words for feelings: Alexithymia as a fundamental personality dimension at the interface of cognition and emotion. *Cognition and Emotion*, *35*(3), 435–448. <https://doi.org/10.1080/02699931.2021.1916442>
- Luminet, O., Nielson, K. A., & Ridout, N. (2021b). Cognitive-emotional processing in alexithymia: An integrative review. *Cognition and Emotion*, *35*(3), 449–487. <https://doi.org/10.1080/02699931.2021.1908231>
- Mckeown, B., Poerio, G. L., Strawson, W. H., Martinon, L. M., Riby, L. M., Jefferies, E., McCall, C., & Smallwood, J. (2021). The impact of social isolation and changes in work patterns on ongoing thought during the first COVID-19 lockdown in the United Kingdom. *Proceedings of the National Academy of Sciences*, *118*(40), e2102565118. <https://doi.org/10.1073/pnas.2102565118>

Mulholland, B., Goodall-Halliwell, I., Wallace, R., Chitiz, L., Mckeown, B., Rastan, A., Poerio, G. L., Leech, R., Turnbull, A., Klein, A., Milham, M., Wammes, J. D., Jefferies, E., & Smallwood, J. (2023). Patterns of ongoing thought in the real world. *Consciousness and Cognition*, *114*, 103530. <https://doi.org/10.1016/j.concog.2023.103530>

Mulholland, B., Chitiz, L., Wallace, R., Mckeown, B., Milham, M., Klein, A., Leech, R., Jefferies, E., Poerio, G., Wammes, J., Stewart, J., Hardikar, S., & Smallwood, J. (2025). Patterns of Ongoing Thought in the Real World and Their Links to Mental Health and Well-Being. *bioRxiv*, 2024.07.22.604681. <https://doi.org/10.1101/2024.07.22.604681>

Myin-Germeys, I., Kasanova, Z., Vaessen, T., Vachon, H., Kirtley, O., Viechtbauer, W., & Reininghaus, U. (2018). Experience sampling methodology in mental health research: New insights and technical developments. *World Psychiatry*, *17*(2), 123–132. <https://doi.org/10.1002/wps.20513>

Poerio, G. L., Klabunde, M., Bird, G., & Murphy, J. (2024). Interoceptive attention and mood in daily life: An experience-sampling study. *Philosophical Transactions of the Royal Society B: Biological Sciences*, *379*(1908), 20230256. <https://doi.org/10.1098/rstb.2023.0256>

Sakuragi, M., Shinagawa, K., Terasawa, Y., & Umeda, S. (2024). The body mirroring thought: The relationship between thought transitions and fluctuations in autonomic nervous activity mediated by interoception. *Consciousness and Cognition*, *125*, 103770. <https://doi.org/10.1016/j.concog.2024.103770>

Smallwood, J., & Schooler, J. W. (2015). The Science of Mind Wandering: Empirically Navigating the Stream of Consciousness. *Annual Review of Psychology*, *66*(1), 487–518. <https://doi.org/10.1146/annurev-psych-010814-015331>

Smallwood, J., Turnbull, A., Wang, H. T., Ho, N. S., Poerio, G. L., Karapanagiotidis, T., ... & Jefferies, E. (2021). The neural correlates of ongoing conscious thought. *Iscience*, *24*, 102132. <https://doi.org/10.1016/j.isci.2021.102132>

Terwee, C. B., Bot, S. D., de Boer, M. R., Van der Windt, D. A., Knol, D. L., Dekker, J., ... & de Vet, H. C. (2007). Quality criteria were proposed for measurement properties of health status questionnaires. *Journal of clinical epidemiology*, *60*(1), 34-42.

Wallace, R. S., Mckeown, B., Goodall-Halliwell, I., Chitiz, L., Forest, P., Karapanagiotidis, T., Mulholland, B., Turnbull, A., Vanderwal, T., Hardikar, S., Gonzalez Alam, T. R., Bernhardt, B. C., Wang, H.-T., Strawson, W., Milham, M., Xu, T., Margulies, D. S., Poerio, G. L., Jefferies, E., ... Smallwood, J. (2025). Mapping patterns of thought onto brain activity during movie-watching. *eLife*, *13*, RP97731. <https://doi.org/10.7554/eLife.97731>

Reviewer 1

Reviewer's Comment	Author Reply
The authors have made thoughtful and thorough revisions based on the reviewer feedback. Additional background information and clarification throughout the manuscript improve the readability and impact of the paper. I am largely satisfied with the revisions made.	We are grateful for the reviewer's constructive feedback, which was invaluable in refining the final version of this paper.
The only point that I have lingering questions about is possible reporting differences associated with alexithymia. In response to my previous point ("Theoretically, participants' level of access and/or awareness of the target of repeated assessment, affect, will vary across the dimension of alexithymia. Do the authors have concerns that this presents a measurement issue?"), the authors performed analyses to rule out "significant differences in reported valence, arousal, and stress across all alexithymia measures". However, I still wonder about aspects of reporting besides response magnitude, such as number of responses, time to respond/time per item or per survey, and also within-person variability in responses on items. These features are commonly examined in EMA research to identify 'careless responding' (Jaso, B. A., Kraus, N. I., & Heller, A. S. (2022). Identification of careless responding in ecological momentary assessment research: From posthoc analyses to real-time data monitoring. Psychological Methods, 27(6), 958–981. https://doi.org/10.1037/met0000312), and the authors indeed used a number of responses and time to response as exclusion criteria. Although there do not appear to be citations for the chosen criteria, which would be helpful. It would also be helpful to see descriptive statistics of these	We thank the reviewer for drawing our attention to this potential confounding factor. Following the reviewer's recommendation, we have further examined response characteristics associated with alexithymia traits. Overall, we did not observe a more careless or restricted response patterns in people with higher alexithymia, despite slightly faster response times. We believe the lack of significant differences in response patterns does not undermine the construct of alexithymia; the trait is primarily defined by difficulties in identifying and interpreting emotions, which may not manifest as reduced engagement or restricted response range in self-report tasks. We have included the following paragraph in the method section and reported the descriptive statistics in the supplementary materials (Supplementary Table 2). “To determine whether our findings were confounded by potential behavioural biases—such as careless or restricted responding—that may be associated with alexithymia, we examined several established measures for identifying abnormal response patterns in experience sampling research⁴³. As shown in Supplementary Table 2, these descriptive indices were largely comparable across low and high alexithymia levels.”

response features for included individuals and observations. In part because these response features can also help us understand -how- people are completing the surveys, beyond what they report. My lingering questions are whether these response features are associated with alexithymia. Are individuals who score higher on alexithymia submitting fewer surveys? responding faster or slower (perhaps indicating degree of deliberation)? or reporting within a more narrow range (e.g. only using 10 pts of a 100 pt scale across all observations)? If so, we should consider if/how that impacts these results and broader affect EMA research. If not, why? If not, doesn't that undermine the definition of alexithymia as a difficulty accessing and identifying feeling states? These seem like important questions to me, but I also do not feel like the manuscript is unsuitable for publication without further investigation. Moreover, alexithymia is not my expertise, thus, I will leave it to the editor's discretion on whether more revision or investigation is necessary. Thank you for the opportunity to review this promising work.	
---	--

Reviewer 2

Reviewer's comment	Author Reply
The authors have addressed all the questions/concerns raised by the reviewer. I do not have further comments.	We appreciate the reviewer's insightful feedbacks, which have greatly improved the clarity and rigor of our manuscript.